# Pericytes regulate VEGF-induced endothelial sprouting through VEGFR1

Hanna M. Eilken[1,4], Rodrigo Diéguez-Hurtado[1], Inga Schmidt[1], Masanori Nakayama[1,2], Hyun-Woo Jeong[1], Hendrik Arf[1], Susanne Adams[1], Napoleone Ferrara[3] & Ralf H. Adams [1]

Pericytes adhere to the abluminal surface of endothelial tubules and are required for the formation of stable vascular networks. Defective endothelial cell-pericyte interactions are frequently observed in diseases characterized by compromised vascular integrity such as diabetic retinopathy. Many functional properties of pericytes and their exact role in the regulation of angiogenic blood vessel growth remain elusive. Here we show that pericytes promote endothelial sprouting in the postnatal retinal vasculature. Using genetic and pharmacological approaches, we show that the expression of vascular endothelial growth factor receptor 1 (VEGFR1) by pericytes spatially restricts VEGF signalling. Angiogenic defects caused by pericyte depletion are phenocopied by intraocular injection of VEGF-A or pericyte-specific inactivation of the murine gene encoding VEGFR1. Our findings establish that pericytes promote endothelial sprouting, which results in the loss of side branches and the enlargement of vessels when pericyte function is impaired or lost.

[1] Max Planck Institute for Molecular Biomedicine, Department of Tissue Morphogenesis and University of Münster, Faculty of Medicine, D-48149 Münster, Germany. [2] Max Planck Institute for Heart and Lung Research, Laboratory for Cell Polarity and Organogenesis, 61231 Bad Nauheim, Germany. [3] University of California San Diego Medical Center, 9500 Gilman Drive, La Jolla, CA 92093, USA. [4] Present address: Bayer AG, Aprather Weg 18a, 42113 Wuppertal, Germany. Hanna M. Eilken and Rodrigo Diéguez-Hurtado contributed equally to this work. Correspondence and requests for materials should be addressed to R.H.A. (email: ralf.adams@mpi-muenster.mpg.de)

Pericytes are vessel-associated (mural) support cells, which belong to the mesenchymal cell lineage and differ substantially from fully differentiated vascular smooth muscle cells (vSMCs), fibroblasts, or other mesenchymal cell types. As there is a lack of strictly pericyte-specific markers, the unambiguous identification of these cells often requires immunostaining of multiple antigens or careful analysis of morphological criteria. Pericytes directly contact capillary endothelial cells (ECs) and both cell types utilize a common basement membrane[1, 2]. Pericytes share certain molecular markers, such as expression of the proteoglycan NG2/Cspg4 or the intermediate filament protein desmin, with vSMCs. The latter, however, cover larger caliber arteries and veins, and are separated by the subendothelial basement membrane from the underlying EC monolayer. Genetic fate mapping experiments in the developing murine heart have established that pericytes and vSMCs are derived from common progenitors and therefore belong to the same cell lineage[3, 4]. While the functional roles of pericytes are currently not fully understood, it is widely accepted that they help to stabilize the vessel wall and prevent vascular leakage. The loss or detachment of pericytes has been implicated in diseases, such as diabetic retinopathy and is also thought to promote cancer metastasis[1, 5]. In the murine brain, pericytes promote establishment of the blood-brain barrier (BBB), which involves the expression of BBB-associated genes and the restriction of vesicular transcytosis in the endothelium[6, 7]. Likewise, loss of pericytes in the murine retina has been recently linked to breakdown of the blood-retina barrier (BRB) and infiltration of inflammatory cells[8, 9]. Based on in vitro co-culture experiments it has been proposed that pericytes and, in particular, their contractility controls EC sprouting and proliferation[10–12]. Pericytes have been also linked to vessel plasticity, regression and thereby patterning of remodeling vascular networks[13].

The recruitment of pericytes in the developing vasculature is mediated by the release of platelet-derived growth factor B (PDGF-B) by ECs, which activates the corresponding receptor, the tyrosine kinase PDGFRβ, on pericytes[14–16]. Accordingly, Pdgfb or Pdgfrb full knockouts or various hypomorphic mutations in these genes lead to strongly reduced pericyte numbers and various vascular defects in embryonic and postnatal mice[16–18]. In particular, Pdgfb or Pdgfrb loss of-function embryos show vascular hyperplasia, microvessel dilation, and upregulation of vascular endothelial growth factor A (VEGF-A) expression[19]. The latter binds and activates the receptor tyrosine kinase VEGFR2 on ECs, which triggers vascular growth and EC proliferation, increases vascular permeability[20, 21] and may explain edema formation in late gestation Pdgfb/Pdgfrb knockout embryos[19]. The expression and activity of VEGF-A during development need to be carefully controlled[22–24]. Signalling through VEGF-A and VEGFR2 is opposed by the receptor VEGFR1/Flt1, another member of the VEGF receptor family, which binds VEGF-A with high affinity but has weak kinase activity and is also produced as a secreted form lacking the cytoplasmic kinase domain[25, 26]. Various studies have established that this antagonistic function of VEGFR1/Flt1 and expression of the receptor by ECs limit vascular growth[27–29]. Loss of Flt1 function results in increased EC proliferation, impaired sprouting and reduced formation of vessel branches, a phenotype that is also seen in the postnatal retinal vasculature after intraocular injection of recombinant VEGF-A[30–32]. Thus, VEGF-A/VEGFR2-induced signalling needs careful regulation to ensure the proper balance between EC proliferation and vessel patterning during the angiogenic expansion of vascular beds.

Here, we have investigated the function of pericytes in the growing postnatal retinal vasculature with inducible genetic experiments in mice. This approach made use of Pdgfrb-CreERT2 transgenic mice, which express tamoxifen-inducible Cre recombinase (CreERT2) in PDGFRβ-expressing cells and therefore preferentially in pericytes and vSMCs of various organs[4, 33]. Acute ablation of these mural cells via CreERT2-controlled diphtheria toxin (DTA) or diphtheria toxin receptor (DTR) expression led to impaired endothelial sprouting and vascular patterning similar to defects seen after intraocular VEGF-A injection or Pdgfrb-CreERT2-mediated inactivation of the murine Flt1 gene. The sum of our data identifies pericytes as important regulators of VEGF signalling and thereby vascular patterning.

## Results

**Inducible pericyte depletion in the postnatal vasculature.** To enable genetic fate mapping and gene inactivation experiments in mural cells in vivo, we have recently developed transgenic mice expressing tamoxifen-inducible CreERT2 recombinase under the control of a large murine Pdgfrb genomic BAC clone[4, 33]. Analysis of these Pdgfrb-CreERT2 mice in the postnatal retina, a well-established model for the characterization of angiogenic blood vessel growth, showed robust recombination of the Rosa26-mTmG Cre reporter[34] (i.e., GFP expression) in mural cells (Supplementary Fig. 1a, b). Similar experiments with NG2-CreER (also known as B6.Cg-Tg(Cspg4-cre/Esr1*)BAkik/J) BAC transgenic mice[35] yielded comparably patchy and inhomogeneous Cre reporter recombination (Supplementary Fig. 1c, d). To validate the specificity of Pdgfrb-CreERT2 mice, we interbred the line with Rpl22tm1.1Psam transgenic animals[36], which enable Cre recombinase-controlled RiboTag analysis of actively translating mRNAs in Cre-positive cells. Following tamoxifen administration from postnatal day (P1) to P3, RT-qPCR analysis of transcripts associated with hemagglutinin (HA) epitope-tagged Rpl22 ribosomal protein showed a strong enrichment of the pericyte markers Pdgfrb and Cspg4/Ng2 relative to input (Supplementary Fig. 1e). In contrast, the endothelial markers Pecam1 and Cdh5 were not enriched in these experiments, which argues against ectopic Pdgfrb-CreERT2 activity in the P6 retinal endothelium.

To address the efficiency of mural cell-specific recombination in retina, we interbred Pdgfrb-CreERT2 and ROSA-DTA transgenic mice[37]. The latter express diphtheria toxin in a Cre-dependent fashion, which enables genetic ablation of specific cell types. Following administration of tamoxifen from P1 to P3, Pdgfrb-CreERT2 ROSA-DTA double transgenic (DTA^iPC) retinas were largely depleted of NG2+, vessel associated cells (Supplementary Fig. 2a–c). Pericyte coverage, as well as gene expression of several mural cell markers in whole-retina lysates were severely affected (Supplementary Fig. 2f, g). Diphtheria toxin-mediated ablation of pericytes was most effective in the peripheral retinal vasculature, whereas some residual NG2+ cells were seen in the central plexus (Supplementary Fig. 2b, c). In addition, this genetic manipulation also led to a strong reduction in arterial vSMC coverage, whereas ectopic αSMA+ immunostaining was seen in scattered cells within the mutant but not littermate control capillary plexus (Supplementary Fig. 2d). Furthermore, mural cell ablation caused reduced outgrowth of the retinal vasculature, vessel dilation and loss of side branches (Supplementary Fig. 2a, e). Together, these experiments establish that Pdgfrb-CreERT2 transgenic mice are specific for pericytes and vSMCs in the postnatal retina, show no detectable Cre activity in ECs, and enable efficient genetic experiments in mural cells.

Pdgfrb-CreERT2 ROSA-DTA double transgenic mice displayed a strong reduction in bodyweight and failed to survive beyond the first week of postnatal life, which is presumably caused by a strong impairment of vessel wall integrity and profound inflammatory responses similar to other models of postnatal pericyte ablation[8, 9]. To circumvent lethality and avoid indirect

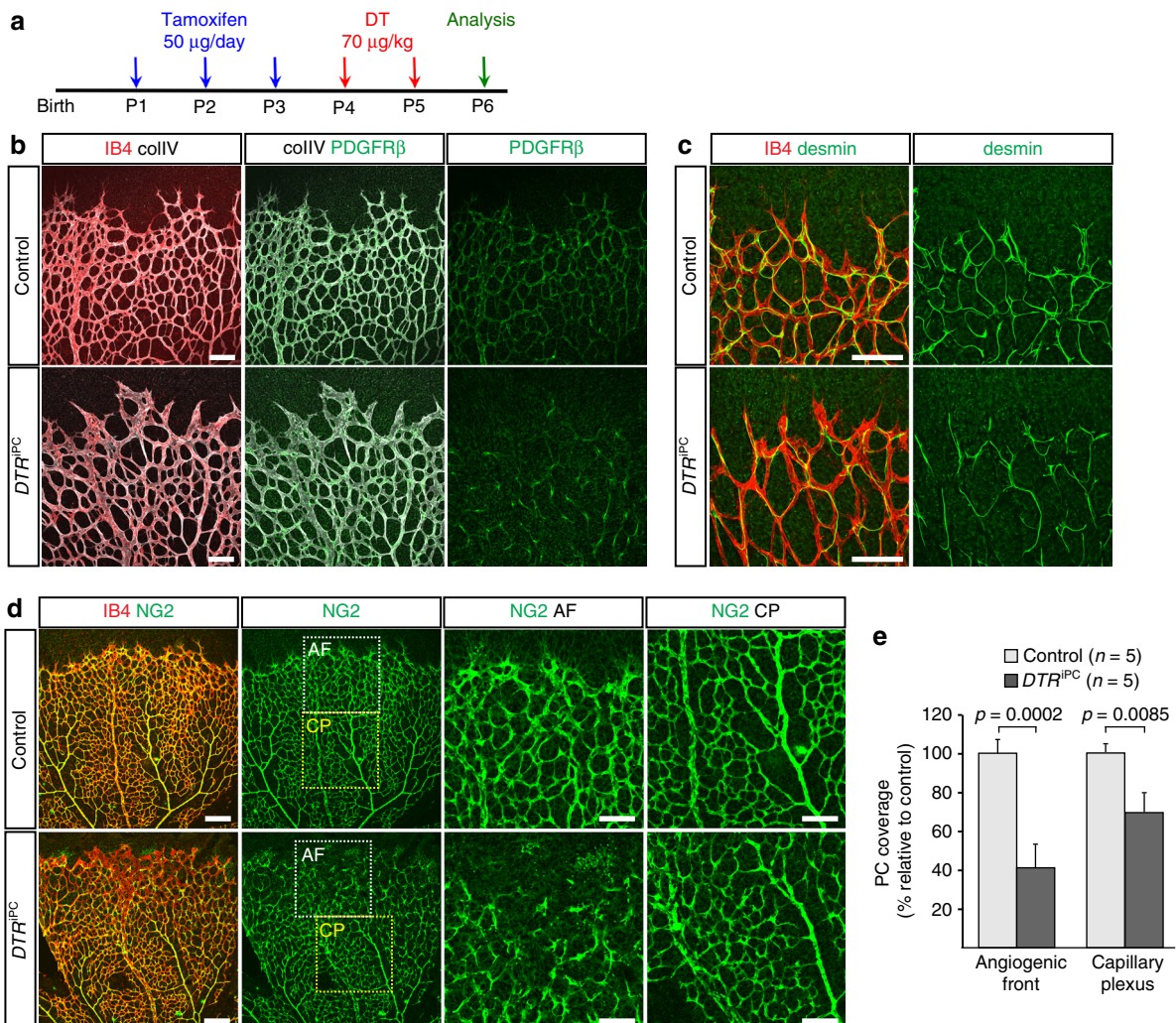

**Fig. 1** Temporally controlled pericyte depletion. **a** Experimental scheme of tamoxifen and diphtheria toxin (DT) administration to postnatal *DTR*iPC mice. **b** Confocal images showing P6 retina whole-mounts stained with isolectin B4 (IB4, red), collagen IV (colIV, white) and PDGFRβ (green). Note reduction of PDGFRβ+ cells at the *DTR*iPC angiogenic front. Scale bar, 100 μm. **c** High-magnification confocal images of the desmin (green) and IB4 (red) stained peripheral vasculature in the P6 *DTR*iPC and littermate control retina. Note extended morphology of desmin+ intermediate filaments in residual *DTR*iPC perivascular cells. Scale bar, 100 μm. **d** Maximum intensity projections of P6 control and *DTR*iPC retinas stained for NG2 (green) and IB4 (red). Images in the two right columns show higher magnifications of insets representing angiogenic front (AF, white dashed squares) and central plexus (CP, yellow-dashed squares), respectively. Scale bar, 200 μm (images on the left) and 100 μm (higher magnifications). **e** Quantitation of pericyte (PC) coverage in the control and *DTR*iPC angiogenic front and capillary plexus, as indicated. Error bars, s.e.m. *p*-values, Student's *t*-test

effects due to profound tissue inflammation, we switched to another strategy aiming at acute and limited pericyte ablation. *ROSA26-iDTR* mice[38] express diphtheria toxin receptor in a Cre-controlled fashion, which makes recombined cells susceptible to ablation following toxin administration, whereas all Cre-negative murine cell types are unaffected because they lack receptor expression. We also limited the dose of administered diphtheria toxin to avoid strong effects on animal health or survival (Methods section). A strategy involving administration of tamoxifen from P1 to P3 followed by two injections of diphtheria toxin (at P4 and P5) led to a significant reduction of PDGFRβ+, NG2+, and desmin+ retinal pericytes in *Pdgfrb-CreERT2 ROSA26-iDTR* double transgenic (*DTR*iPC) mice at P6 (Fig. 1a±d). The depletion of pericytes was more effective in the angiogenic front region, i.e., around actively growing capillaries, than in the more mature central vessel plexus (Fig. 1d, e).

**Regulation of EC proliferation and sprouting by pericytes.**
Following diphtheria toxin/DTR-mediated pericyte ablation,

bodyweight, and radial outgrowth of retinal vessels were reduced in *DTR*iPC mice relative to control littermates (Fig. 2a–c). Pericyte depletion also led to a significant reduction of branch points in the retinal vasculature, whereas there was no increase in vessel pruning, as indicated by control-like numbers of empty (collagen IV+, isolectin B4−) collagen sleeves (Fig. 2d–f). Immunosignals for the extracellular matrix (ECM) proteins fibronectin and collagen IV, and the active form of integrin β1, a subunit contributing to several integrin family heterodimeric cell-matrix receptors and an important regulator of blood vessel growth and integrity[39], were also comparable in pericyte-ablated and control retinal vessels (Supplementary Fig. 3a–c). Whereas other models of pericyte ablation compromised the integrity of endothelial cell-cell junctions[8, 9], the more limited reduction of pericytes in *DTR*iPC mice did not affect the localization of the junction proteins VE-cadherin, Claudin 5 and Zonula occludens-1 (ZO1) (Supplementary Fig. 3d–f). In contrast, the number of endothelial sprouts and the extension of filopodia, a feature that is usually associated with the leading tip cells at the distal end of

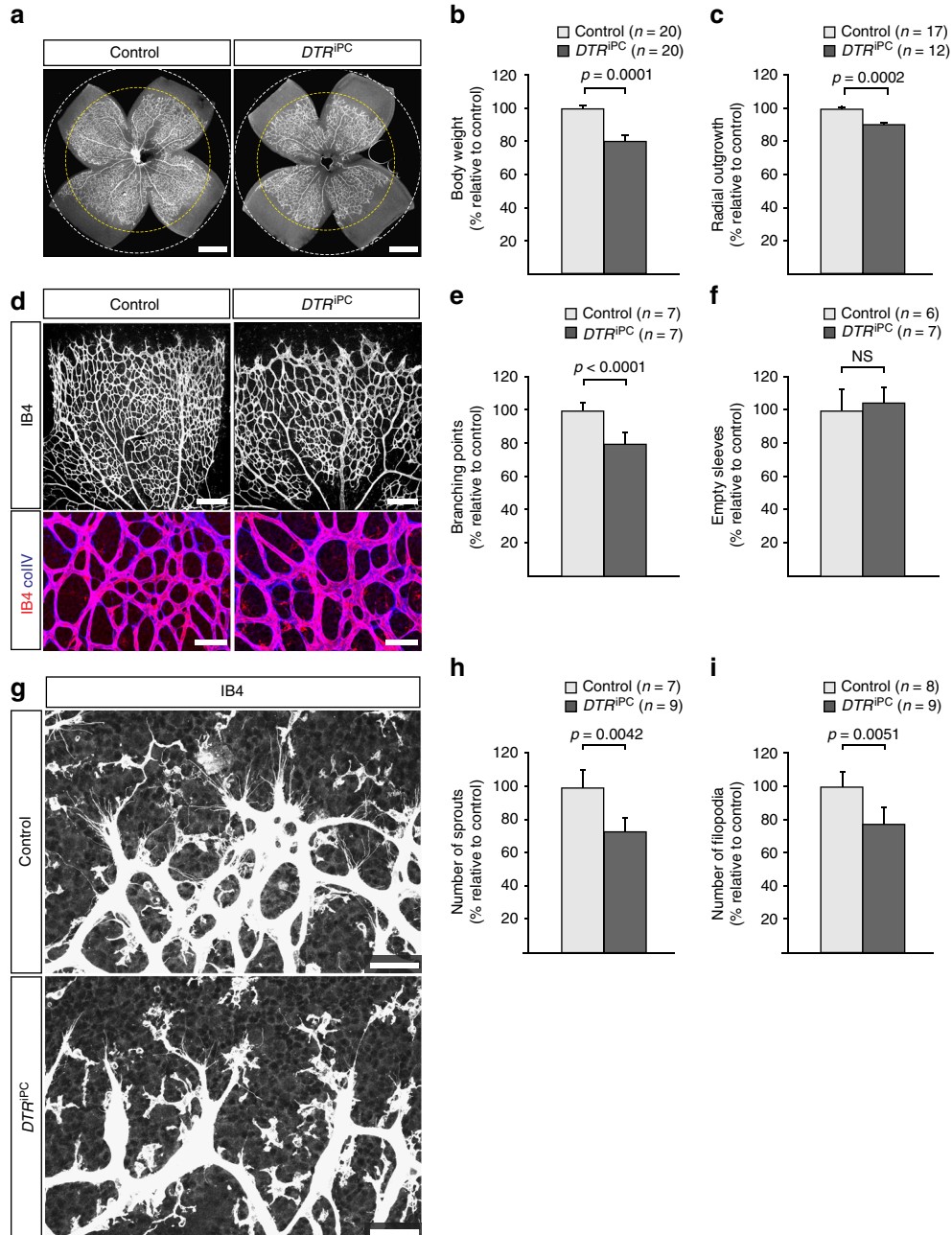

**Fig. 2** Altered endothelial sprouting after pericyte depletion. **a** Isolectin B4-stained P6 control and *DTR*[iPC] retina whole-mounts. Dashed circles indicate vessel-covered (yellow) and peripheral avascular (white) areas, respectively. Scale bar, 500 μm. **b, c** Quantitation of P6 control and *DTR*[iPC] body weight **b** and radial outgrowth of the retinal vasculature **c**. Error bars, s.e.m. *p*-values, Student's *t*-test. **d** Confocal images of IB4 (white/red) and collagen IV (colIV, blue) showing the reduction of branch points but no increase in IB4- colIV+ empty matrix sleeves in *DTR*[iPC] P6 retinas. Scale bar, 200 μm (top panels) and 50 μm (bottom). **e, f** Quantitation of branch points **e** and empty matrix sleeves **f** in P6 retinas. Error bars, s.e.m. *p*-values, Student's *t*-test. NS, not statistically significant. **g** Maximum intensity projection of the IB4-stained P6 control and *DTR*[iPC] retinal angiogenic front illustrating differences in sprout number, morphology and filopodia number after pericyte depletion. Scale bar, 50 μm. **h, i** Quantitation of sprouts **h** and filopodia number **i** in the P6 control and *DTR*[iPC] angiogenic front. Error bars, s.e.m. *p*-values, Student's *t*-test

sprouts[30, 40], were significantly reduced after pericyte depletion (Fig. 2g–i).

While overall EC proliferation was slightly but not significantly altered, diphtheria toxin/DTR-mediated pericyte ablation led to the regional increase of EdU+ ECs at the leading edge of the angiogenic growth front. This was accompanied by a slight but statistically not significant reduction in endothelial proliferation within the vessel plexus (Fig. 3a–c). Consistent with the reduced outgrowth of the retinal vasculature, the local density of EC nuclei

(identified by expression of the transcription factor Erg1) per area in the peripheral plexus was increased after pericyte depletion (Fig. 3d). Moreover, the ablation of pericytes led to profound morphological changes in endothelial sprouts at the leading edge of the growing vascular plexus. While control sprouts were slender and contained 1–2 EC nuclei, *DTR*[iPC] sprouts were thick, blunt-ended and consisted of large EC clusters (Fig. 3e). These clusters enclosed an enlarged luminal area, as indicated by immunostaining for the apical marker ICAM2 (Fig. 3f).

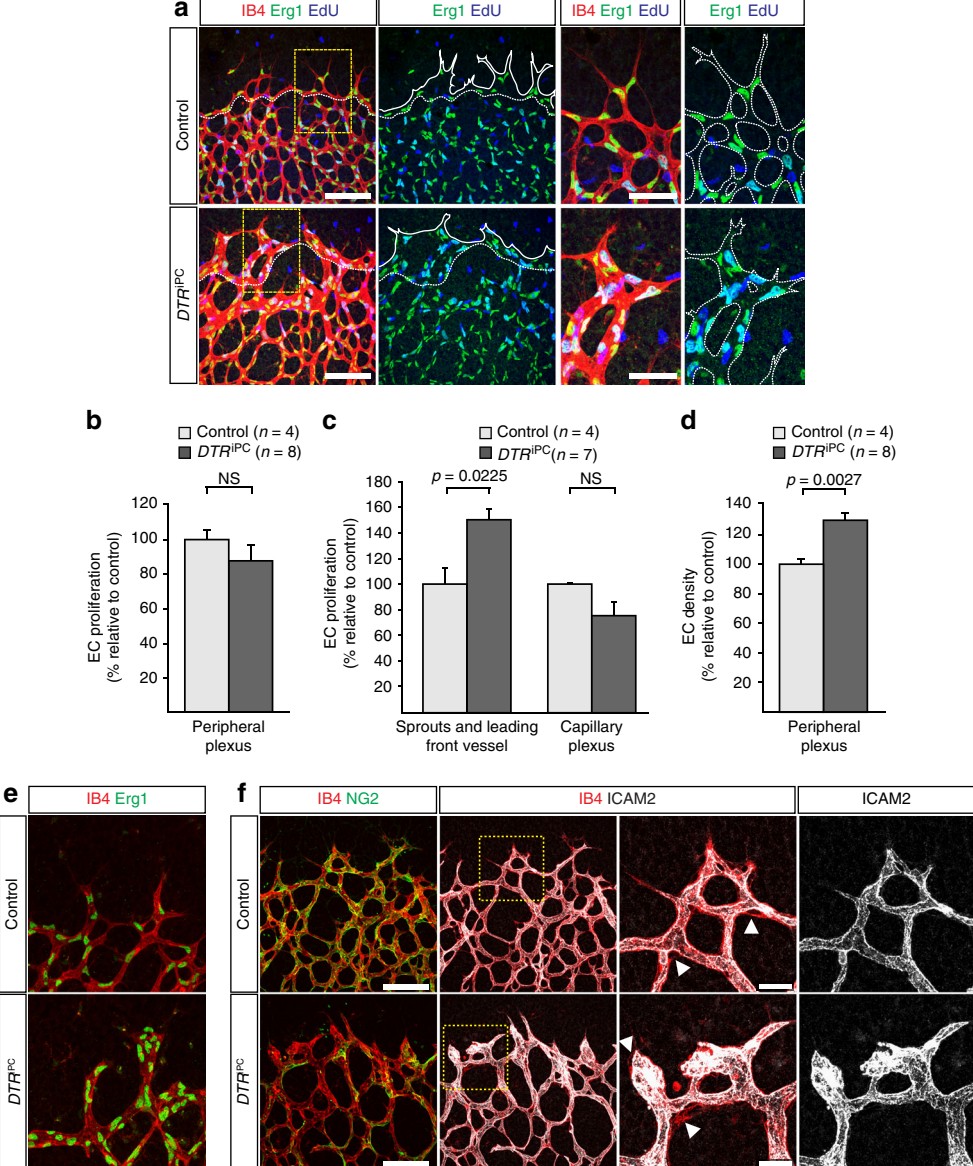

**Fig. 3** Pericytes control endothelial behavior in angiogenesis. **a** Confocal images showing proliferating cells (EdU, blue), IB4 (red) and EC nuclei (Erg1, green) in the P6 control and *DTR*iPC retinal angiogenic front, as indicated. Images on the right show higher magnification of boxed insets in left column. Dashed lines outline the angiogenic leading vessel from where the sprouts emerge (second column) or vessel shape (fourth column) at the edge of the vascular plexus. Scale bar, 100 μm (left panels) and 50 μm (higher magnifications). **b**–**d** Quantitation of total **b** and regional EC proliferation **c** as well as EC density **d** within the peripheral vascular plexus of P6 control and *DTR*iPC retinas. Error bars, s.e.m. *p*-values, Student's *t*-test. NS, not statistically significant. **e** High magnification of IB4 (red) and Erg1 (green) stained sprouts highlighting the accumulation of EC nuclei in P6 *DTR*iPC but not control sprouts. Scale bar, 50 μm. **f** Confocal images showing IB4 (red), NG2+ pericytes (green) and vessel lumen (ICAM2, white) in the control and *DTR*iPC angiogenic front of P6 retinas. Images on the right show higher magnification of boxed insets in the second column. Note that IB4 signal covers a larger area than the apical ICAM2 immunostaining (arrowheads in third column). Scale bar, 100 μm (left panels) and 50 μm (higher magnifications)

As the VEGF-A/VEGFR2 signalling pathway is an important regulator of EC sprouting and proliferation, we next investigated whether diphtheria toxin/DTR-mediated pericyte ablation might lead to changes in this pathway. In contrast to recently reported mouse models with very extensive pericyte loss[8, 9] or our own *DTA*iPC model (Supplementary Fig. 4e) of genetically controlled diphtheria toxin expression, which all show a strong upregulation of VEGF-A expression indicative of tissue hypoxia and inflammation, the pattern and intensity of VEGF-A immunostaining was not altered in *DTR*iPC mutants (Fig. 4a, g). However, immunosignals for VEGFR2 were enhanced in the thick *DTR*iPC sprouts at the angiogenic front (Fig. 4b, h). Likewise, immunostaining of the receptor VEGFR3 was increased at the *DTR*iPC

angiogenic growth front (Fig. 4c, i), whereas Tie2, a receptor tyrosine kinase binding angiopoietin family ligands[41], failed to show enrichment in sprouts (Fig. 4d). The expression of VEGFR3 is positively regulated by VEGFR2 signalling[42, 43] arguing that the activity of the latter might be enhanced in *DTR*iPC sprouts. Indeed, expression of dermatan sulfate proteoglycan endocan/ Esm1 (endothelial cell-specific molecule 1), which is positively regulated downstream of VEGF-A/VEGFR2[44, 45], was increased in the *DTR*iPC angiogenic front (Fig. 4e, j). Expression of Dll4, a ligand for Notch family receptors and critical regulator of angiogenic EC behavior, is also positively regulated by VEGFR2 signalling[46, 47]. Dll4 distribution was highly heterogeneous in *DTR*iPC sprouts (Fig. 4f), which either lacked Dll4

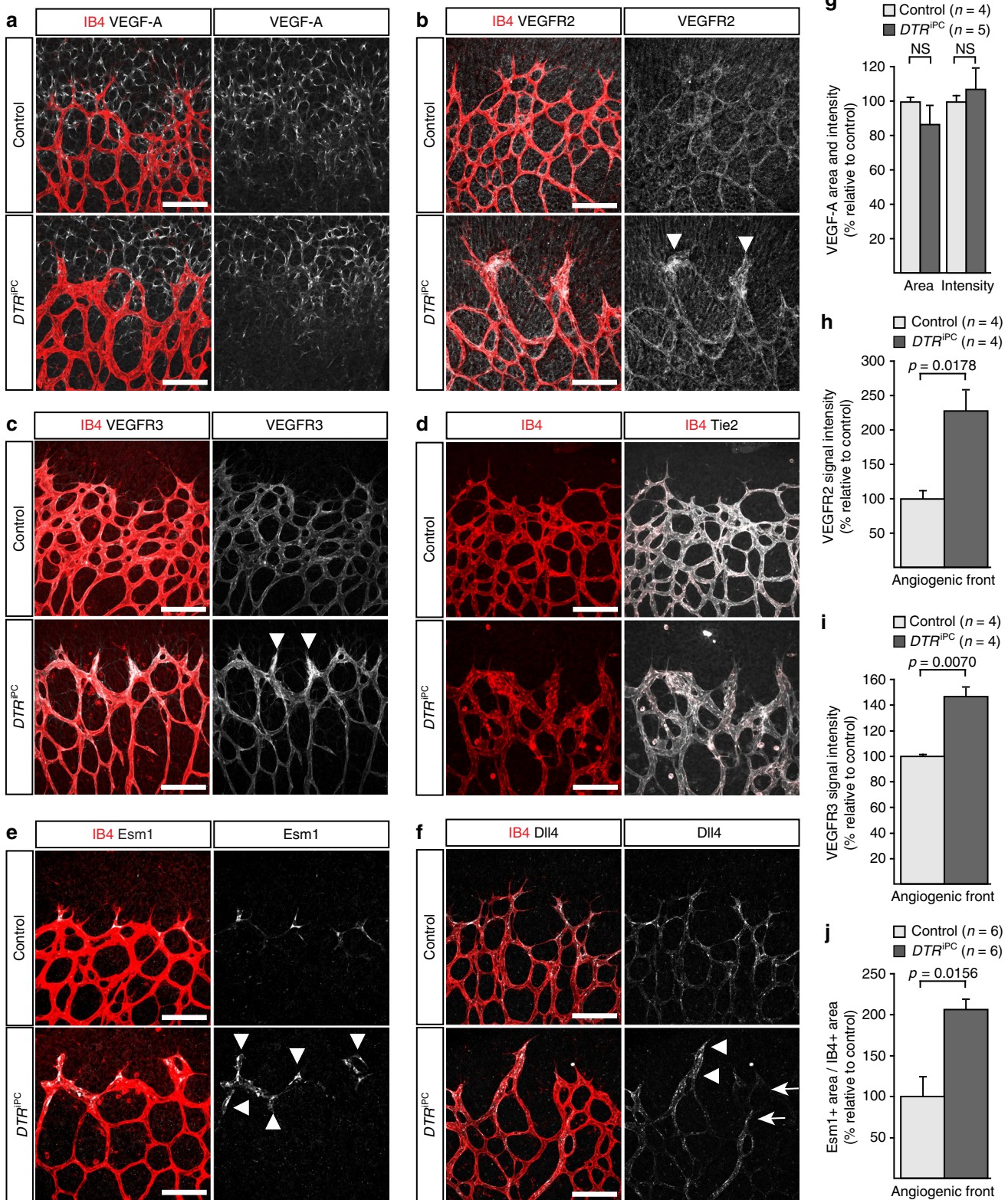

**Fig. 4** Endothelial changes after pericyte depletion. **a–f** Maximum intensity projection of confocal images from control and *DTR*[iPC] P6 retinas stained for IB4 (red) in combination with VEGF-A **a**, VEGFR2 **b**, VEGFR3 **c**, Tie2 **d**, Esm1 **e**, and Dll4 **f** (all in white), as indicated. Note local increase of VEGFR2, VEGFR3, and Esm1 (arrowheads in **b**, **c**, **e**) but not Tie2 or VEGF-A at the edge of the vessel plexus. Dll4 expression in *DTR*[iPC] sprouts is increased in some regions (arrowheads) but absent in others (arrows). Scale bar, 100 μm. **g-j** Quantitation of VEGF-A immunosignals area and intensity **g**, signal intensity for VEGFR2 **h** and VEGFR3 **i** and proportion of Esm1+ area with respect to vascular area **j** in the P6 control and *DTR*[iPC] angiogenic front. Error bars, s.e.m. *p*-values, Student's *t*-test

immunosignals or showed ectopic expression of the ligand that extended into the sprout base and the adjacent capillary plexus. Such a pattern of Dll4 expression has been previously proposed to reflect enhanced and asynchronous VEGF-A/VEGFR2 activity[48]. Upregulation of VEGFR2 and Esm1 expression was also observed in $DTA^{iPC}$ retinas (Supplementary Fig. 4a–d). Likewise, elevated

VEGFR2 expression in the cortex of P6 $DTA^{iPC}$ brains suggests that pericytes control endothelial VEGFR2 levels in multiple tissues (Supplementary Fig. 4f–i).

**VEGF-dependent effects on EC proliferation and sprouting.** Administration of exogenous VEGF-A or genetically encoded

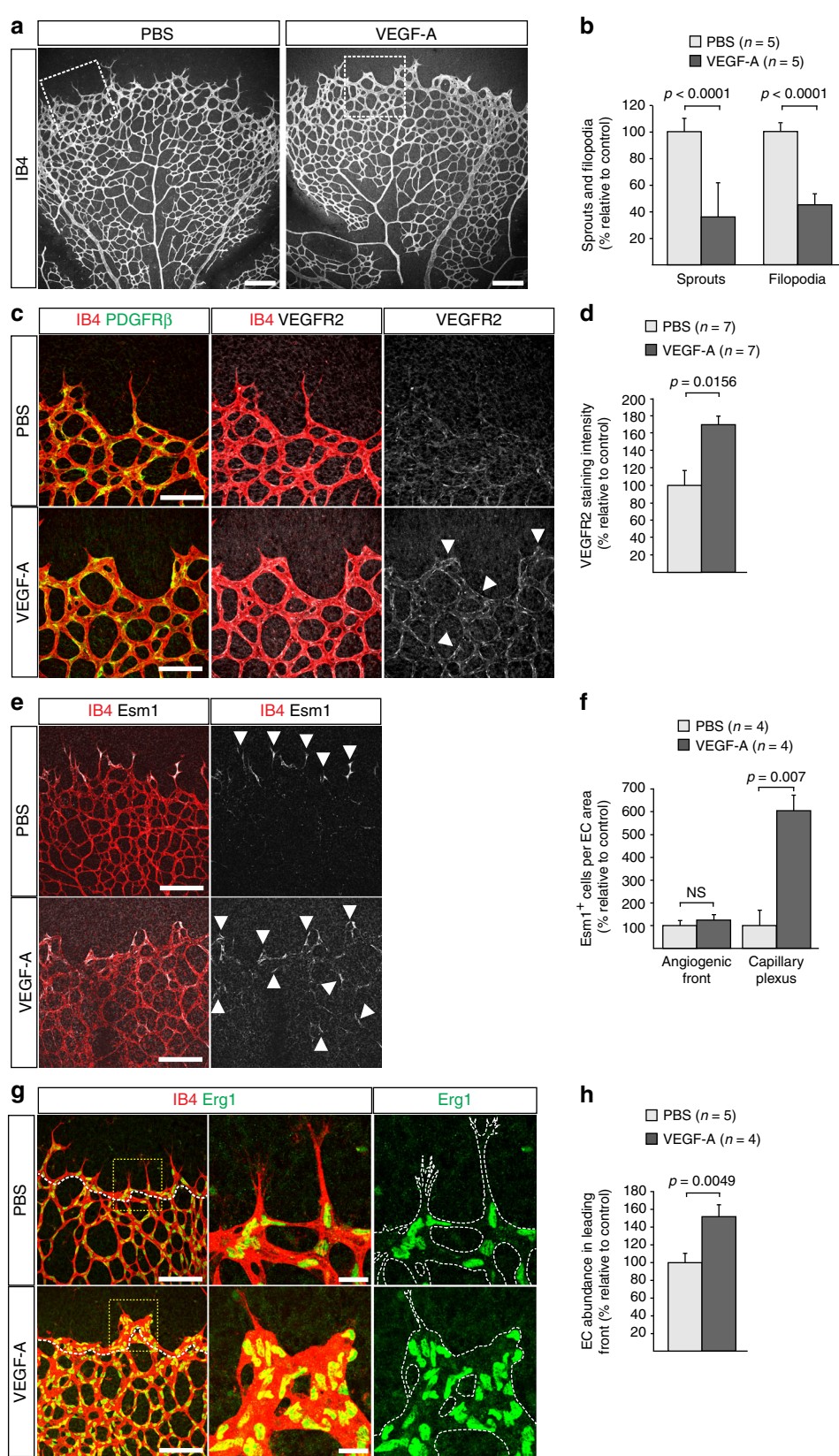

selective expression of the soluble VEGF-A120 isoform, which lacks matrix-binding and thereby the ability to provide spatially restricted cues, led to short and dilated vessel sprouts[30, 40] similar to those seen after pericyte depletion (Figs. 1b, 2g). To directly investigate the effect of ectopic VEGF signalling on the organization of vessel sprouts and regional EC proliferation, we administered human VEGF-A165 or vehicle control to wild-type P6 pups by intraocular injection (Methods section). Analysis of the retinal vasculature at 4 h post-injection revealed that VEGF-A165 treatment did not lead to acute alterations in the distribution of extracellular matrix proteins or loss of PDGFRβ+ or NG2+ pericytes (Supplementary Fig. 5a–g). In contrast, VEGF-A165 treatment induced a significant dilation of peripheral retinal capillaries (Supplementary Fig. 5a, f), which was not caused by altered EC proliferation (Supplementary Fig. 6b, c) but was accompanied by significant reductions in endothelial sprouting

and filopodia extension (Fig. 5a, b, Supplementary Fig. 6a). VEGFR2 and Esm1 immunostaining in the peripheral retina were also increased after VEGF-A165 administration (Fig. 5c–f), an effect that is consistent with the observed increase in *KDR* and *ESM1* transcription after VEGF-A stimulation of HUVECs in vitro (Supplementary Fig. 5h, i). As in *DTR*iPC retinas, sprout-like structures appeared dilated and the VEGF-A165-treated angiogenic front contained an excessive number of ECs relative to vehicle-injected controls (Fig. 5g, h).

The results above argued that pericytes might antagonize VEGF-A/VEGFR2 signalling in the developing vasculature in a previously unknown fashion. As it has been reported that pericytes express the receptor VEGFR1[49–52], we investigated whether pericytes in the postnatal retina are also a source of this receptor and, in particular, its soluble isoform (sFlt1), which is well known to antagonize VEGF-A/VEGFR2 signalling[27–29]. We

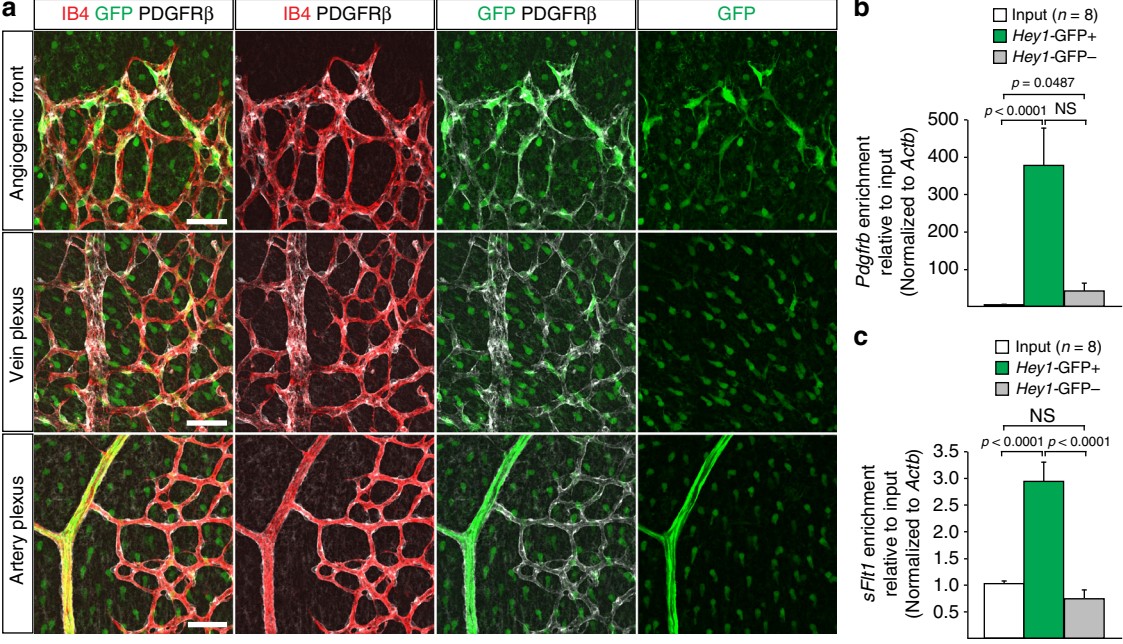

**Fig. 6** Expression of sFlt1 in pericytes at the angiogenic front. **a** Maximum intensity projections of confocal images from P6 retinas of the *Hey1*-GFP transgenic reporter mouse model stained for GFP (green), PDGFRβ+ (white) and IB4 (red). Images of the first row show enrichment of *Hey1*-GFP+, PDGFRβ+ perivascular cells in the angiogenic front in comparison to mural cells covering the remodeling central plexus around veins (middle row) and arteries (bottom row). Note strong expression of *Hey1*-GFP reporter in arterial ECs (bottom row). Scale bar, 50 μm. **b**, Quantitation of *Pdgfrb* expression by qPCR in P6 PDGFRβ+ retinal pericytes sorted based on GFP expression in comparison to whole-retina single-cell suspension (input). Note significant enrichment of *Pdgfrb* in both (GFP+ and GFP−) pericyte fractions and higher expression in the *Hey1*-GFP+ subset. Error bars, s.e.m. *p*-values, Kruskal–Wallis and Dunn's multiple comparison test. NS, not statistically significant. **c** Quantitation of sFlt1 expression by qPCR in sorted P6 retinal pericytes in comparison to whole-retina single-cell suspension (input). Note significant enrichment of sFlt1 expression in *Hey1*-GFP+ pericytes in comparison to input and GFP- pericytes. Error bars, s.e.m. *p*-values, one-way ANOVA and Tukey's multiple comparison test. NS, not statistically significant

**Fig. 5** Vascular alterations after intraocular VEGF-A injection. **a** Morphology of IB4-stained P6 wild-type retinal vessels at 4 h after administration of human VEGF-A165 (0.5 μl at a concentration of 5 μg μl⁻¹). Note blunt appearance of the vessel front after VEGF-A injection but not for vehicle (PBS) control. Scale bar, 200 μm. **b** Quantitation of sprouts and filopodia at the front of the P6 vessel plexus after injection of VEGF-A165 or vehicle control. Error bars, s.e.m. *p*-values, Student's *t*-test. **c** PDGFRβ+ (green) pericytes are unaffected by short-term VEGF-A administration, whereas VEGFR2 immunosignals (white) are increased in IB4+ (red) ECs (arrowheads). Images shown correspond to insets in **a**. Scale bar, 100 μm. **d** Quantitation of VEGFR2 immunosignals intensity in the peripheral plexus of P6 retinas after injection of VEGF-A165 or vehicle control. Error bars, s.e.m. *p*-values, Student's *t*-test. **e** Confocal images showing increased Esm1 immunostaining (white) in IB4+ (red) ECs in the peripheral plexus (arrowheads) after VEGF-A injection in P6 pups. Scale bar, 200 μm. **f** VEGF-A165 injection-mediated increase of Esm1 immunosignals (normalized to IB4+ EC area) in the peripheral capillary plexus but not at the edge of the angiogenic front in comparison to PBS-injected controls at P6. Error bars, s.e.m. *p*-values, Student's *t*-test. NS, not statistically significant. **g** Short-term VEGF-A165 administration leads to clustering of Erg1+ (green) and IB4+ (red) ECs, as indicated, in thick sprout-like structures of P6 retinas. Panels in the center and on the right (scale bar, 20 μm) show higher magnification of the insets on the left (scale bar, 100 μm). Dashed lines in panels on the right outline IB4+ vessels. **h** Quantitation of EC density in the leading front vessel and emerging sprouts of the P6 angiogenic front after injection of VEGF-A165 or vehicle control. Error bars, s.e.m. *p*-values, Student's *t*-test

also made use of *Hey1-GFP* (*Tg(Hey1-EGFP)ID40Gsat*) reporter mice (http://www.gensat.org/), which preferentially label the presumably more immature pericytes near the angiogenic front and express much higher levels of *Pdgfrb* relative to total PDGFRβ+ retinal pericytes (Fig. 6a, b). Because of their close proximity to the leading edge of the growing retinal vasculature (Fig. 6a), the subset of *Hey1-GFP*-expressing pericytes is more likely to directly control endothelial sprouting than the majority of pericytes, which cover capillaries of the central retina. Indeed, RT-qPCR analysis of *Hey1-GFP*+ PDGFRβ+ cells showed significant enrichment of sFlt1 expression relative to input and

also relative to *Hey1-GFP*-negative PDGFRβ+ cells (Fig. 6c). Notably, pericytes at the angiogenic front were also preferentially ablated in *DTR*iPC mutant retinas (Fig. 1b–e).

Next, we studied the effect of *Pdgfrb-CreERT2*-controlled *Flt1* inactivation (*Flt1*iPC) in the postnatal vasculature (Fig. 7a). Whereas *Flt1*iPC mutants displayed no changes in viability, bodyweight and vascular outgrowth, sprouting and endothelial filopodia formation were defective (Fig. 7b–e). Indicating that the dosage of pericyte-specific VEGFR1 expression is critical, a similar phenotype was seen in *Flt1*iPC/+ heterozygotes (Fig. 7b–e). Both heterozygous and homozygous mutants showed

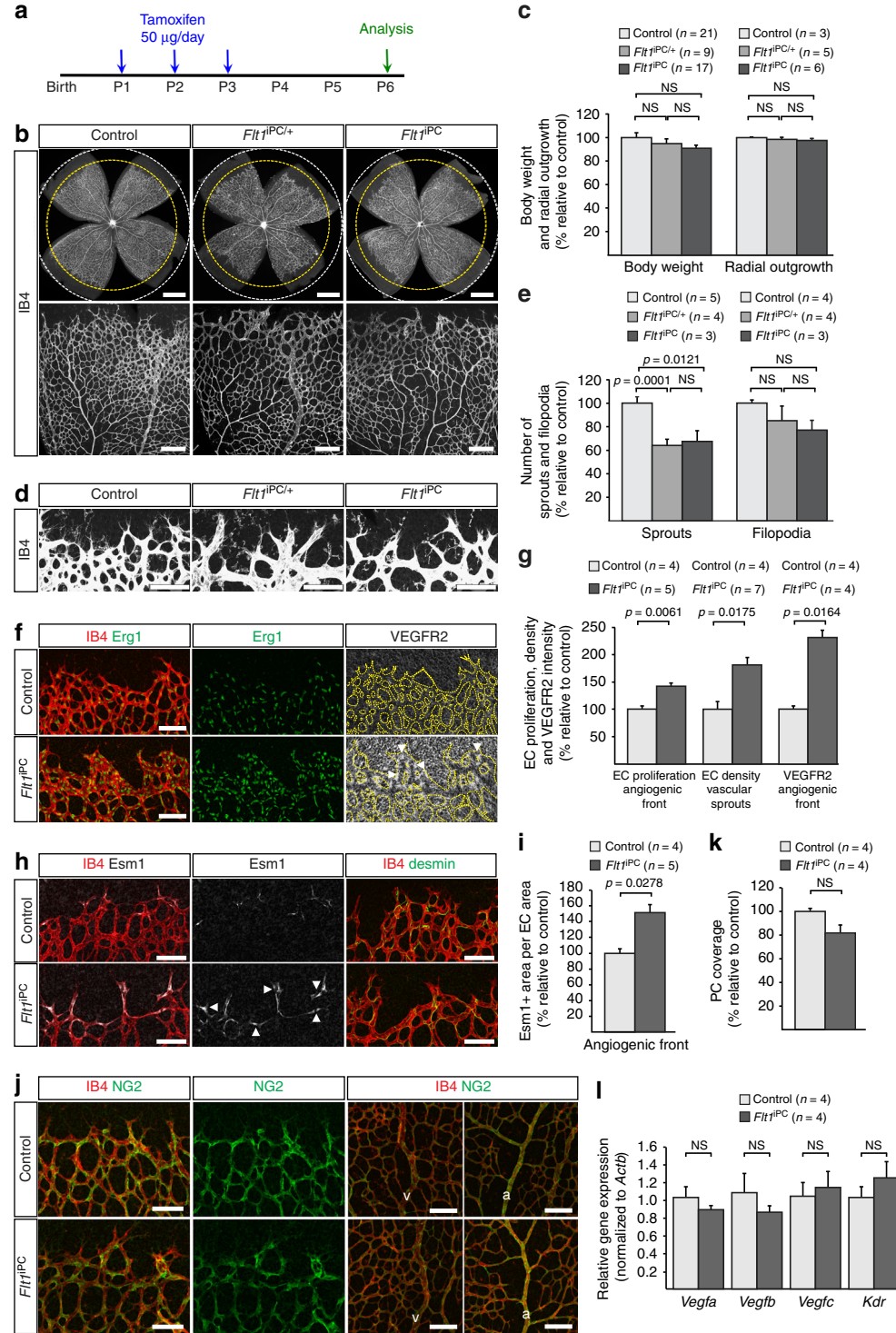

pronounced thickening of sprouts, the local accumulation of total and proliferating ECs at the angiogenic front, and upregulated expression of VEGFR2 and Esm1 (Fig. 7d, f–i, Supplementary Fig. 7a, b). At the same time, the number of pericytes was only slightly but not significantly reduced in the Flt1iPC/+ and Flt1iPC retinal vasculature (Fig. 7h, j, k, Supplementary Fig. 7c, d). Global expression of transcripts encoding components of the VEGF signalling (Vegfa, Vegfb, Vegfc, Kdr) or the angiopoietin signalling (Angpt1, Angpt2, Tek) pathways was not significantly altered in Flt1iPC retina lysates relative to littermate control samples, whereas expression of Pdgfrb was increased (Fig. 7l, Supplementary Fig. 7e, f). We also analyzed the role of VEGFR1 expression by pericytes in a later stage of retinal development, namely the perpendicular sprouting from the superficial vascular plexus into the deeper retina (Supplementary Fig. 7g). This approach revealed profoundly reduced vascularization of the P10 Flt1iPC deeper retina without overt changes in the coverage by NG2+ pericytes (Supplementary Fig. 7h–k). Inactivation of Flt1 in PDGFRβ+ pericytes also led to reduced expansion of the vasculature and endothelial sprouting in the postnatal (P6) brain cortex, which was accompanied by upregulated VEGFR2 immunostaining (Supplementary Fig. 8a–g). Together, these data establish that Pdgfrb-CreERT2-mediated inactivation of Flt1 phenocopies key aspects of the defects seen after pericyte ablation, which argues that pericytes control VEGF-A activity at the edge of the growing retinal vasculature but also within other tissues, such as the brain.

To compare the relevance of EC-specific vs. pericyte-specific VEGFR1 expression, we also used Cdh5-CreERT2 transgenic mice[53] to inactivate Flt1 in the postnatal endothelium. The resulting Flt1iEC mutants showed no overt reductions in radial outgrowth of the retinal vasculature or pericyte abundance (Supplementary Fig. 9a–d). However, mutants displayed statistically significant defects in vessel branching, vascular area, sprout formation and EC proliferation (Supplementary Fig. 9e–i). Whereas Esm1 was visibly upregulated in the Flt1iEC angiogenic front, sprouts did not show the same morphological alterations as seen in pericyte-specific mutants (Fig. 7b, d, Supplementary Fig. 9f). Taken together, these results indicate that EC behavior at the edge of the growing vasculature is controlled by VEGFR1 from multiple cellular sources. Pericytes emerge as important regulators of endothelial sprouting and branch formation, which is, at least in part, mediated by the modulation of VEGF-A/VEGFR2 signalling activity.

## Discussion

It is widely appreciated that pericytes stabilize growing vessels and control endothelial permeability[1, 2], but their exact role in vascular growth and patterning remains little understood. In Pdgfb or Pdgfrb full knockouts, which are perinatal lethal due to renal, cardiovascular, placental, and hematological defects[15, 16], the chronic lack of pericytes leads to capillary dilation and the formation of microaneurysms in the embryonic brain but no changes in vessel branching[19]. In contrast, constitutive mutants lacking the PDGF-B retention motif, a sequence of basic amino acid residues mediating binding to heparan sulfate polysaccharide chains, displayed impaired endothelial sprouting and the reduction of branch points[18]. A complication of constitutive mutants is the strong interdependence of placental, cardiac and vascular development[54, 55], which makes it difficult to distinguish primary and secondary, potentially indirect morphogenetic alterations. Nevertheless, the phenotype of PDGF-B retention motif mutants is similar to the vascular malformations seen in DTAiPC and DTRiPC mice. In particular, acute pericyte ablation in postnatal mice leads to profound alterations in the behavior of angiogenic ECs, which accumulate in thick protrusions and fail to form the normal number of sprouts seen in control littermates. This impaired sprouting is also the likely cause of the reduction in branch points, whereas we failed to see evidence for increased vessel pruning in the pericyte-depleted retinal vasculature. However, this does not rule out that pericytes may regulate vascular remodeling in other organs or developmental stages, as has been previously suggested[13].

Various pathways have been implicated in the communication between ECs and pericytes. While PDGF-B/PDGFRβ and TGFβ signalling are thought to control processes such as pericyte recruitment, proliferation and differentiation[14, 56–58], much less is known about the regulation of ECs by pericyte-derived signals. Such a role has been attributed to angiopoietin-1, a secreted ligand on the receptor tyrosine kinase Tie2 in ECs[41]. However, this conclusion is based on in vitro experiments and pharmacological studies in wild-type mice[59–61], whereas the actual function of pericyte-derived angiopoietin-1 remains to be investigated. Other factors that have been proposed to mediate the crosstalk between pericytes and ECs include Notch family receptors and their ligands, the cell adhesion molecule N-cadherin and heparan sulfate proteoglycans[62–66]. The latter category includes molecules, such as syndecans and NG2, which are expressed by pericytes and thereby may modulate signalling by heparan sulfate-associated VEGF and other growth factors. Upregulated VEGF-A expression, which has been observed in Pdgfb/Pdgfrb knockout embryos[19], as well as in other models of pericyte ablation[8, 9], is likely to contribute to the EC hyperplasia after the loss of pericytes. Pericytes also express the receptor VEGFR1 and both VEGF-A and the related placental growth factor (PlGF) have been shown to induce pericyte ablation in the mature

**Fig. 7** Inactivation of Flt1 in PDGFRβ+ cells. **a** Experimental scheme of tamoxifen administration for the generation of Flt1iPC mutants. **b** P6 control, Flt1iPC/+ and Flt1iPC retinas stained with isolectin B4 (IB4). Dashed circles indicate vessel-covered (yellow) and peripheral avascular (white) areas in the overview pictures (top). Scale bar, 500 μm. **c** Quantitation of body weight and radial outgrowth of the retinal vasculature in control, Flt1iPC/+ and Flt1iPC P6 pups. Error bars, s.e.m. p-values, one-way ANOVA. NS, not statistically significant. **d** Confocal images of the IB4-stained P6 control, Flt1iPC/+ and Flt1iPC retinal angiogenic front illustrating differences in sprout number and morphology. Scale bar, 100 μm. **e** Quantitation of sprouts and filopodia in P6 control, Flt1iPC/+ and Flt1iPC retinas. Error bars, s.e.m. p-values, one-way ANOVA and Tukey's multiple comparison test. NS, not statistically significant. **f** Confocal images of IB4 (red), Erg1 (green) and VEGFR2 (white) stained P6 retinas highlighting the accumulation of EC nuclei and enhanced VEGFR2 immunosignals (arrowheads) in Flt1iPC sprouts. Vessels are outlined by dashed lines on the right panel. Scale bar, 100 μm. **g** Quantitation of EC proliferation (EdU+ Erg1+) at the angiogenic front, EC density in sprouts and leading front vessel and VEGFR2 immunosignals intensity in the angiogenic front of control and Flt1iPC P6 retinas. Error bars, s.e.m. p-values, Student's t-test. **h** Esm1 (white) expression (arrowheads) in the angiogenic front (IB4+, red, first two columns) and detection of desmin+ pericytes (green, third column) in P6 control and Flt1iPC retinas. Scale bar, 100 μm. **i** Quantitation of Esm1+ proportion relative to vascular area (IB4+) in the angiogenic front of control and Flt1iPC P6 retinas. Error bars, s.e.m. p-values, Student's t-test. **j** Confocal images of P6 retinas stained for NG2 (green) and IB4 (red) showing no significant changes in pericyte coverage in the front (first two columns) or the remodeling plexus around veins (v) or arteries (a) (last two columns). Scale bar, 100 μm. **k, l** Quantitation of pericyte coverage **k** and relative gene expression by qPCR on whole lysates **l** in control and Flt1iPC P6 retinas. Error bars, s.e.m. p-values, Student's t-test. NS, not statistically significant

vasculature[49]. While our own data show that *Pdgfrb-CreERT2*-mediated inactivation of *Flt1* only modestly affects pericyte number during development, it turns out that pericytes limit local VEGF signalling by expressing VEGFR1. Our data also indicate that pericytes are molecularly heterogeneous during development and the *Hey1-GFP+* subset, which is characterized by higher soluble VEGFR1 expression than other pericytes, may be particularly relevant for the regulation of sprouting ECs. Interestingly, the role of pericytes in sprouting angiogenesis in the retina is not compensated by endothelial VEGFR1 expression, which is by itself also important for normal developmental angiogenesis, as has been previously shown[27, 28, 31]. Thus, pericytes are important regulators of endothelial angiogenic behavior, which involves the modulation of local VEGF signalling.

## Methods

**Mutant mice and inducible genetic experiments.** *Pdgfrb(BAC)-CreERT2*[4, 33] or *B6.Cg-Tg(Cspg4-cre/Esr1\*)*[BAkik/J] (NG2-CreER[TM]) BAC transgenic mice[35] were interbred with *Rosa26-mTmG* Cre reporter mice[34] or Rosa26-YFP Cre reporter mice[67] in order to evaluate recombination efficiency and specificity. Moreover, *Pdgfrb(BAC)-CreERT2* transgenic mice were interbred with *Rpl22*[tm1.1Psam] animals[36] for RiboTag quantitative analysis of gene expression in Cre-recombined cells of the double transgenic animals.

For pericyte ablation experiments, *Pdgfrb(BAC)-CreERT2* transgenic animals were interbred *with ROSA-DTA*[68] transgenic mice or with *Rosa26-iDTR* mice[38].

For pericyte-specific deletion of VEGFR1, *Pdgfrb(BAC)-CreERT2* transgenic mice were bred with *Flt1* conditional knockout mice[69] to obtain *Pdgfrb(BAC)-CreERT2*[+/T] *Flt1*[+/lox] mice which were bred to *Flt1*[lox/lox] animals in order to obtain litters with control (Cre-negative), heterozygous (*Flt1*[iPC/+]) and homozygous (*Flt1*[iPC]) *Flt1* conditional knockouts. For endothelial-specific deletion of VEGFR1, a similar approach was followed using *Cdh5(PAC)-CreERT2* transgenic animals[53]. All experiments in wild-type animals were conducted on C57Bl/6 mice.

For analysis of specific pericyte subpopulations we made use of heterozygous *Hey1-GFP* (Tg(Hey1-EGFP)ID40Gsat) reporter mice (http://www.gensat.org/) which were maintained by mating with C57Bl/6 mice. Offspring was genotyped by evaluation of GFP epifluorescence.

Cre-mediated recombination was induced by intraperitoneal injection of 50 µg tamoxifen (Sigma, T5648; dissolved in ethanol-peanut oil 1:40 at 1 mg ml$^{-1}$) every 24 h during three consecutive days starting at postnatal day 1 (P1). Pericyte-depletion mediated by diphtheria toxin in *Pdgfrb(BAC)CreERT2; Rosa26-iDTR* mice (DTR[iPC]) was achieved by intraperitoneal injection of diphtheria toxin (Sigma, D0564; 70 µg kg$^{-1}$ dissolved in PBS at 4 µg ml$^{-1}$) on two consecutive days (P4 and P5) after tamoxifen administration. Phenotypes were analyzed at P6.

In all genetic experiments, control animals refer to tamoxifen or diphtheria toxin-injected littermates that do not harbor a Cre transgene. Both males and females were used in this study and they are represented in a 1:1 ratio.

All experiments involving animals were performed in compliance with the relevant laws and institutional guidelines, following protocols previously approved by local animal ethics committees and conducted with permission granted by the Landesamt für Natur, Umwelt und Verbraucherschutz (LANUV) of North Rhine-Westfalia to the Max Planck Institute for Molecular Biomedicine.

**Immunohistochemistry.** Specific conditions of fixation were used depending on the structures to be imaged. In all cases, both control and mutant samples were processed in parallel under identical conditions. Briefly, whole eye globes were fixed in 2–4% paraformaldehyde (PFA, Sigma, P6148) at 4 °C or room temperature (RT) up to 4 h. Eyes were washed with PBS and retinas were dissected and partially cut in four quadrants. Blocking and permeabilizing was achieved by incubation of the retinas in 1% bovine serum albumin (BSA, Sigma, A4378) and 0.3% Triton X-100 (Sigma, T8787) in PBS for 2 h at RT or 4 °C overnight. Primary antibodies were diluted in blocking/permeabilization solution and incubated overnight at 4 °C. After staining with primary antibodies, retinas were washed three times (20 min at RT each) in PBS and counterstained with suitable species-specific secondary antibodies or streptavidin coupled to Alexa Fluor dyes (1:400, all from Invitrogen) for 2 h at RT in blocking/permeabilization buffer. After secondary antibody incubation, retinas were washed three times (20 min at RT each) in PBS and mounted in Fluoromount G (Southern Biotech, 0100-01).

For imaging the blood vessels, staining with biotin-conjugated isolectin B4 (IB4, 1:25, Vector Laboratories, B-1205) was performed. Moreover, the following primary antibodies were used: rabbit anti-collagen type IV (colIV, 1:100, Millipore, AB756P), rat monoclonal anti-laminin α5 (LAMA5, 1:50, clone 4G A2 11, kindly provided by Prof. Dr. Lydia Sorokin, Institute of Physiological Chemistry and Pathobiochemistry, University of Münster, Germany), goat anti-PDGFRβ (1:100, R&D, AF1042), rat anti-PDGFRβ (1:100, eBioscience, 14–1402), rabbit anti-desmin (1:100, Abcam, ab15200), rabbit anti-NG2 (1:200, Millipore, AB5320), rabbit anti-Erg1 (1:100, Abcam, ab110639), rat anti-ICAM2 (1:100, Pharmingen, 553326), rabbit anti-VEGF (1:100, Santa Cruz, sc-152), rat anti-VEGFR2 (1:100,

Pharmingen, 555307), rabbit anti-VEGFR2 (1:100, Cell Signaling, 2479), goat anti-VEGR3 (1:100, R&D, AF743), goat anti-Tie2 (1:100, R&D, AF762), goat anti-Endocan (Esm1, 1:100, R&D, AF1999), goat anti-Dll4 (1:100, R&D, AF1389), mouse anti-αSMA-Cy3 (1:200, Sigma, C6198), rabbit anti-fibronectin (1:100, Sigma, F3648), rat anti-VE-cadherin (1:100, BD Bioscience, 555289), rabbit anti-Claudin-5 (1:100, Zymed, 341600), rat anti-integrin β1 (1:100, BD Biosciences, 553715), rabbit anti-Zonula occludens 1 (ZO1, 1:100, Invitrogen, 617300), chicken anti-green fluorescent protein (GFP, 1:300, AVES Labs, Inc., GFP-1020) and rabbit anti-GFP-Alexa Fluor 488 (1:400, Invitrogen, A21311).

For immunohistochemical analysis of mouse brains, pups at postnatal day 6 (P6) were anesthetized by intraperitoneal injection of xylazine (Bayer, Rompun 2%; 10 mg kg$^{-1}$) and ketamine (Zoetis, Ketavet 100 mg ml$^{-1}$; 100 mg kg$^{-1}$) dissolved in saline. After opening the thoracic cavity and exposing the heart, the right atrium was pierced and 10 ml of PBS were perfused through the left ventricle using a peristaltic pump (Pump P1, GE Healthcare, BioSciences AB, 18-1110-91) in order to wash out the blood from the major circulatory system. Right after, 10 ml of 4% PFA (Sigma, P6148) at 4 °C were perfused in the same manner to start fixation of the tissues. Brains were carefully dissected out from the skull and postfixed for up to 8 h by immersion in 4% PFA at 4 °C, then washed in PBS and sagittal mid-line sectioned using an acrylic brain matrix for mouse (World Precision Instruments, RBMA-200C). Individual brain hemispheres were embedded in 55 °C low gelling temperature agarose (Sigma, A9414, 4% in PBS) and immediately placed on top of ice to cool down. Once solidified, agarose blocks were trimmed, glued to a specimen holder with cyanoacrylate (UHU GmbH and Co. KG) and 100 µm sections were obtained using a vibratome (Leica, VT 1200 S). Vibratome sections were blocked and permeabilized by overnight incubation in 1% BSA (Sigma, P6148) and 0.5% Triton X-100 (Sigma, T8787) in PBS at 4 °C. Primary antibodies were diluted in blocking/permeabilization solution supplemented with 2% normal donkey serum (Abcam, ab7475) and incubated overnight at 4 °C. After staining with primary antibodies, vibratome sections were washed three times (20 min at 4 °C each) in 0.3% Triton X-100 in PBS and incubated overnight with suitable donkey-raised, species-specific, Alexa Fluor-coupled secondary antibodies (all from Invitrogen) diluted (1:400) in blocking/permeabilization buffer with 2% normal donkey serum. After secondary antibody incubation, vibratome sections were washed twice in 0.3% Triton X-100 in PBS (20 min at 4 °C each) and once in PBS before mounting in Fluoromount G (Southern Biotech, 0100-01).

The following primary antibodies were used for immunostaining of mouse brain vibratome sections: rabbit anti-GLUT1 (1:200, Millipore, 07-1401), rat anti-PDGFRβ (1:100, eBioscience, 14–1402), rat anti-ICAM2 (1:200, Pharmingen, 553326), rabbit anti-VEGFR2 (1:100, Cell Signaling, 2479) and goat anti-VEGR2 (1:100, R&D, AF644).

**Proliferation assay in vivo.** For in vivo analysis of cell proliferation, pups were injected intraperitoneally with 5-ethynyl-2-deoxyuridine (EdU, Life Technologies, A10044; 30 mg kg$^{-1}$ dissolved in DMSO-PBS 1:10 at 2 mg ml$^{-1}$) 2 h before dissection. Retinas were dissected and stained as described before. After secondary antibody staining, EdU labeling was detected by means of a Click-it EdU Alexa Fluor-647 Imaging Kit (Life Technologies, C10340) according to the manufacturer's instructions.

**VEGF-A intraocular injections.** Intraocular injection of human recombinant VEGF-A[165] (Reliatech, 300-076) was performed in P6 pups that were anesthetized by intraperitoneal injection of xylazine (Bayer, Rompun 2%; 10 mg kg$^{-1}$) and ketamine (Zoetis, Ketavet 100 mg ml$^{-1}$; 100 mg kg$^{-1}$) dissolved in saline. The eyelids were carefully separated using a scalpel and 0.5 µl of VEGF-A[165] at a concentration of 5 µg µl$^{-1}$ were injected into the vitreous humor using glass capillary pipettes with a micromanipulator (Nanoject II, Drummond Scientific). Pups were kept under controlled conditions 2 h before EdU injection. Samples were collected 2 h after EdU administration (i.e., 4 h after VEGF-A[165] injection).

**VEGF-A in vitro stimulation.** Passage 4 human umbilical vein endothelial cells (HUVECs, Thermo Fisher, C0035C) were seeded in 24-well CellBIND plates (Corning, 3337) at 5000 cells per cm$^2$ density and cultured in EBM-2 endothelial growth basal medium (Lonza, CC-3156) supplemented with 2% fetal bovine serum (FBS, Lonza, CC-410A). Twenty-four hours after seeding, cells were stimulated with the same media used for culture supplemented with 200 pM of human recombinant VEGF-A[165] (Reliatech, 300-076), which was prepared freshly every day. After 1, 2, 3, or 4 days of stimulation, cells were washed once with D-PBS (Sigma, D8537) and lysed in 400 µl of RLT Plus buffer (Qiagen) supplemented with β-mercaptoethanol (10 µl ml$^{-1}$). RNA isolation and cDNA synthesis was performed as described later in this section. The results from three independent experiments, each consisting of at least three individual wells per condition, were averaged for the final quantitation.

**Image acquisition, processing, and analysis.** Stained and flat mounted retinas were imaged at high resolution with a Leica TCS SP5 confocal microscope equipped with the following objective lenses: ×10 HC PL APO Numerical Aperture (NA) 0.4, ×20 HCX PL S-APO NA 0.5, ×40 HCX PL APO NA 1.25, and ×63 HCX PL APO NA 1.40.

The cortex from stained brain vibratome sections was imaged with a Zeiss LSM880 confocal laser scanning microscope equipped with the following objective lenses: ×10 Plan Apo NA 0.45, ×20 Plan Apo NA 0.80, ×40 C Apo NA 1.20, and ×63 Plan Apo NA 1.40.

Low magnification, whole-retina pictures were acquired with a MZ16F stereomicroscope (Leica) coupled to a digital camera (Hamamatsu, C4742-95).

Image acquisition, analysis and processing was performed using LAS-AF 2.6 (Leica), ZEN Software 2.3 SP1 (Carl Zeiss), Volocity 6.3 (Perkin Elmer), Photoshop CS6 13.0 (Adobe), Illustrator CS6 16.0.0 (Adobe) and Fiji-IJ 2.0.0-rc-43[70] software. All confocal images shown where immunostaining levels are compared or quantified are representative of 6–12 different images from three or more replicate experiments (animals) per group. Settings for scanner confocal detection and laser excitation were always kept identical between samples whenever comparisons between mutant mice and their respective controls were done.

**Quantitative analysis of retina and brain vasculature**. In all quantitations shown the control group was set as reference (100%) and the corresponding value for the mutant group was calculated accordingly. For all the quantitative analysis based on confocal microscopy images, averages obtained from a minimum of 2–6 comparable fields of view per sample were used. The number of animals per group used in every experiment is indicated in each figure.

Radial outgrowth, a quantitative measure for retinal vascularization, was calculated by dividing the length of the retinal area covered by blood vessels by the total retinal length (both measures starting from the optical nerve).

Pericyte coverage was calculated by dividing the surface area positive for pericyte marker immunolabeling (i.e., NG2 for retinas and PDGFRβ for brains or retinas) into the surface area positive for endothelial marker (i.e., IB4 for retinas and GLUT1 for brains). For calculating the number of pericytes per vascular area, individual pericytes were manually counted using desmin staining and normalized to the vascular area identified by IB4 staining.

Branching points were manually counted and normalized to the vascular area in each field. Likewise, pruning events were manually counted after visual identification of colIV+ IB4– empty matrix sleeves and normalized to the vascular area identified by IB4 staining.

Quantitation of sprouts and filopodia was performed in high magnification (×40 or ×63) pictures from hard-fixed retinas. Visual identification of sprouts was based on the following criteria: IB4+ vascular projections emerging from/connected to the distal edge vessels and extending filopodia towards the avascular space. Filopodia were identified as slender cytoplasmic projections extending beyond the leading edge of sprouts. The number of sprouts and filopodia in each field of view quantified was normalized to the corresponding length of the angiogenic front measured by drawing a line running along the base of all emerging sprouts. Likewise, quantitation of sprouts in the vasculature of the brain's cortex was performed by manual counting of filopodia-emitting endothelial (GLUT1+) projections from already established vessels and normalized to the vascular area in each field of view quantified.

For quantitation of EC density and proliferation, the EC-specific nuclear marker (Erg1) was used for unambiguous identification. Proliferating ECs (EdU+ and Erg1+) were visually identified, manually counted, and their number normalized to the total amount of Erg1+ ECs and the vascular area (IB4+).

For quantitation of VEGFR2 and VEGFR3 immunoreactivity the vascular area was identified and segmented based on IB4+ staining. Within this area, the intensity of VEGFR2 or VEGFR3 immunosignals was quantified for every pixel and averaged. Next, a second intensity quantitation was performed for the respective VEGFR2 or VEGFR3 channels in the non-vascular area (i.e., the original image after cropping out all areas identified as IB4+). The signal intensity plotted corresponds to the value obtained after subtracting the non-vascular mean intensity from the vascular-specific mean intensity for each pixel quantified.

For quantitation of VEGF-A immunosignals area and intensity a similar approach was used, nevertheless, in this case object selection and segmentation for quantitative analysis was based on VEGF-A+ immunosignals. Normalization was done by subtraction of the background staining (mean intensity for VEGF-A in the original images after cropping out the VEGF-A+ area).

For quantitation of the relative abundance of Esm1+ ECs, retinas from P6 pups perfused with PBS and 4% PFA (as described previously) were used in order to reduce the background staining associated to red blood cells and serum. Briefly, Esm1+ immunosignals were detected and the surface area they represent was quantified and normalized to the whole vascular area identified by IB4 immunostaining.

When quantitations were limited to the angiogenic front, a 260 μm-wide region of interest (ROI) starting from the most distal tip cells was drawn enclosing the peripheral vascular plexus and following the outline of the angiogenic front.

**Fluorescence activated cell sorting of retinal pericytes**. Eyeballs from *Hey1-GFP* (*Tg(Hey1-EGFP)ID40Gsat*) reporter mice were collected in MEM (Gibco, 31095-029) supplemented with 25 mM HEPES (PAA, S11-001), penicillin/streptomycin (Gibco, 15140) and 10% fetal calf serum. Retinas were immediately dissected and enzymatically digested using the Papain Dissociation System (Worthington Biochemical Corporation, LK003150). The single-cell suspension was filtered using a 50 μm cell strainer (Sysmex, 04-0042-2317), washed with FACS buffer (2% fetal calf serum and 2 mM EDTA pH 8.0 in PBS), centrifuged for 5 min at 300×g and resuspended in 200 μl of FACS buffer with the following antibodies:

rat anti-CD140b-APC (1:25, eBiosciences, 17-1402-82), rat anti-TER-119-Pacific Blue (1:200, BioLegend, 116232), rat anti-CD45-Pacific Blue (1:200, BioLegend, 103126), rat anti-CD31-PerCP/Cy5.5 (1:50, BioLegend, 102420) and rat anti-CD140a-PE/Cy7 (1:100, eBiosciences, 25–1401). After 40 min of incubation at 4 °C, the samples were washed with 1 ml of FACS buffer, centrifuged for 5 min at 300×g and resuspended in a final volume of 400 μl FACS buffer with DAPI (1 μg ml$^{-1}$, Sigma D9542). A 20 μl aliquot of the whole-retina single-cell suspension was kept as input and lysed in 350 μl of RLT Plus buffer (Qiagen) supplemented with β-mercaptoethanol (10 μl ml$^{-1}$). Cell sorting was performed with a FACS Aria IIu (BD Biosciences) using a 70 μm nozzle. Gating strategies involved positive selection of live, single cells based on DAPI exclusion, as well as forward and side-scatter parameters. CD45, TER-119, CD31, and CD140a (PDGFRα) were used as negative selection markers in order to deplete hematopoietic lineages, red blood cells, endothelial, and mesenchymal cells, respectively. CD140b (PDGFRβ) was used for positive selection of putative pericytes and within this fraction GFP+ and GFP– cells were directly sorted to RLT Plus buffer (Qiagen) supplemented with β-mercaptoethanol (10 μl ml$^{-1}$). Each sample was derived from two mice (4 pooled retinas).

**Polysome-bound mRNAs isolation using RiboTag**. Immunoprecipitation and purification of polysome-bound mRNAs was performed from dissected retinas, snap frozen in liquid nitrogen and stored at −80 °C. Sample preparation involved disruption of the tissue using pellet pestles (749540-0000; Kimble Chase) in 400 μl of polysome buffer (50 mM Tris, pH 7.5, 100 mM KCl, 12 mM MgCl$_2$, 1% Nonidet P-40, 1 mM DTT, 200 U/ml RNasin, 1 mg/ml heparin, 100 μg/ml cyclohexamide and 1× protease inhibitor) and centrifugation at 20,000×g for 10 min at 4 °C. Before immunoprecipitation, a 10 μl aliquot of the supernatant was kept as input; the remaining volume was transferred to a new tube, mixed with 25 μl of magnetic beads conjugated to an anti-HA-tag antibody (MBL, M180-11) and incubated on an orbital shaker at 4 °C overnight. The magnetic beads were washed four times with high-salt buffer (50 mM Tris, pH 7.5, 300 mM KCl, 12 mM MgCl$_2$, 1% Nonidet P-40, 1 mM DTT, and 100 μg/ml cyclohexamide) and resuspended in 350 μl of RLT Plus buffer (Qiagen) supplemented with β-mercaptoethanol (10 μl ml$^{-1}$).

**RNA isolation, cDNA synthesis, and gene expression analysis**. Whole-retina lysates were prepared by mechanical dissociation of freshly dissected retinas in 350 μl of RLT Plus buffer (Qiagen) supplemented with β-mercaptoethanol (10 μl ml−1). Alternatively, retinas were snap frozen in liquid nitrogen right after dissection and stored at −80 °C before processing. Total RNA from sorted pericytes, HA-tag-immunoprecipitates, whole-retina lysates or cultured HUVECs was extracted using RNeasy Plus Micro Kit (Qiagen, 74034) according to manufacturer's instructions. Analysis of RNA samples was performed using the RNA 6000 Pico Kit (Agilent Technologies, 5067–1513) in a 2100 BioAnalyzer (Agilent Technologies, G2938C). Reverse transcription into cDNA was achieved using the iScript cDNA Synthesis Kit (Bio-Rad, 170–8890) according to manufacturer's instructions. Quantitative PCR (qPCR) was performed on a CFX96 Touch Real-Time PCR Detection System (Bio-Rad) using the SsoAdvanced Universal Probes Supermix (Bio-Rad, 172–5280) and the TaqMan gene expression probes (Thermo Fisher) detailed on Supplementary Table 1. *Actb* or *GAPDH* were used as reference genes.

**Statistical analysis**. All data are presented as mean ± standard error of the mean (s.e.m.). No statistical methods were used to predetermine sample size; instead sample number was defined according to previous experience and reproducibility of the results across several independent experiments. For all experiments done the number of animals (n) analyzed per group was derived from at least two different litters. No animals were excluded from the analysis and no randomization or blinding was used. *P*-value < 0.05 was considered to be statistically significant. Normal distribution was assumed by default unless sample size allowed specific interrogation of normality using the D'Agostino-Pearson omnibus test.

Unpaired two-tailed Student's *t*-test was used to determine statistical significance when comparing two independent groups. For multiple comparisons, one-way ANOVA and Tukey's post hoc test or Kruskal–Wallis and Dunn's post hoc test were used depending on data distribution. Statistical analysis was performed using GraphPad Prism 6 (GraphPad, version 6.0b).

**Data availability**. All relevant data supporting the results of the present study are included within the article, its Supplementary Information files and can be obtained from the corresponding author on reasonable request.

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

## Acknowledgements

Funding was provided by the Max Planck Society, the University of Münster and the Deutsche Forschungsgemeinschaft (FOR 2325, SFB 1009, and Cells-in-Motion Cluster of Excellence; EXC-CiM). H.M.E. was partially funded by the DFG priority program SPP 1190 (The tumor-vessel interface) and R.D.-H. by the European Union (Marie Curie ITN VESSEL). We are grateful to Dr. Martin Stehling for expert advice in flow cytometry experiments and to Dr. Ann Seynhaeve for the contribution to the initial characterization of *Pdgfrb-CreERT2* founder animals.

## Author contributions

H.M.E. and R.D.-H. contributed equally to this work and either has the right to list himself first in bibliographic documents. H.M.E., R.D.-H., I.S., H.A., H.-W.J. and R.H.A. designed experiments and interpreted results. H.M.E., R.D.-H. and R.H.A. wrote the manuscript. H.M.E., I.S. and R.D.-H. performed all in vivo experiments, H.-W.J. did the RiboTag analysis, M.N. the intraocular injections, S.A. and N.F. generated or provided critical genetic tools.

## Additional information

**Competing interests:** The authors declare no competing financial interests.

