## [Peer Review File · Nature Communications]

Reviewers' comments:

Reviewer #1 (Remarks to the Author):

The manuscript by Eilken et al. develops and validates a new transgenic line to evaluate the role of pericytes in vascular development, *Pdgfrb-CreERT2* mice. This line is used to examine the loss of pericytes in the developing post-natal retinal vasculature, and this loss leads to changes in endothelial cell morphogenesis, sprouting and proliferation. Some of these changes are recapitulated in two manipulations that affect VEGF-A signaling: acute intraocular injection of VEGF-A or pericyte deletion of the negative regulator of VEGF-A signaling VEGFR1. The conclusion is that pericytes and pericyte-derived VEGFR1 are important for post-natal retinal morphogenesis. The work is timely in that, while it is clear that pericyte-endothelial cell (EC) interactions are important for vessel morphogenesis, the molecular pathways are not well-defined, and the role of VEGF-A signaling regulation from the PC compartment is not well-characterized. Enthusiasm is somewhat moderated by the focus of the work on characterization of the overall loss of PC via expression of the DTA receptor and toxin injection, relative to characterization of PC-specific loss of VEGFR1, which is considered more novel. Additionally, the work does not carefully explore (or even discuss) the effects of VEGF-A manipulation on PC-EC interactions, which would strengthen the somewhat modest conclusions.

Major issues:

1. The focus on effects of overall PC loss in post-natal retinal development lacks some novelty, as classic papers (Hellstrom et al, 2001; Lindblom et al, 2003) document reduced sprouting and hyperplasia/increased vessel diameter in mice lacking pericytes in the postnatal retina. The focus on effects at the vascular front with somewhat cursory statements regarding effects of PC loss behind the front seems to miss an opportunity to contrast effects in angiogenic vs. remodeling/stabilized areas where PC function is better understood. Although the authors document that PC loss is not complete behind the front, some parameters (i.e. proliferation) have opposite trends depending on the location, and this is not explained or discussed. This group has published beautiful work describing how trafficking that affects VEGF-A signaling is spatially localized in EC in the retina - how might this information overlap with the new PC data? As presented the new work on global PC loss seems to primarily confirm previous work.
2. The analysis of PC loss of VEGFR1 function is novel and intriguing but somewhat incomplete: a single set of data points is presented, predominantly focused on effects at the vascular front. The data set does not completely overlap with the data presented for intraocular VEGF-A injection, making comparisons difficult.
3. The concept that PC interactions with EC may be affected by any of the manipulations is not addressed. This is especially important for the VEGF manipulations that do not ablate PC. PC and EC share a basement membrane - is this intact with VEGF-A manipulation? Where are PC relative to sprouts/filopodia at the front and does this change with VEGF manipulation?
4. Very little discussion of the interesting data that is presented - for example, why would loss of PC decrease branch points? Where do the extra EC come from 4h post-intraocular VEGF-A, since proliferation is not changed (as expected for this short time period)?
5. Some of the data is not quantified, and in some cases the conclusions in the paper are not well-supported by the panels shown, especially given the increased number of EC in mutants.
--Fig 4b, c, e - more EC at front in mutant, stronger if quantified and normalized to EC area
--Fig 5c - more EC at front in mutant where more R2.
--Fig 5d, e - stated in text that more staining of ESM at front, yet graph shows no difference at front, rather more behind the front.
--Fig 6f - R2 not quantified.

--Fig 6h - ESM not quantified and differences in mutant not apparent.

--Fig 6i - desmin seems reduced at front relative to control in presented panels

Minor issues:

1. Fig. 3a - here and other places the definition of the vascular front is unclear. In methods says 260 micrometers from distal most sprout, but here looks much less and the control looks more "tight" than the mutant. since many of the measurements compare this specific area and show differences, further clarification is needed here.

2. Fig. 3e - ICAM-2 does not appear apical in these panels. Suggest perfusion labeling to confirm expanded luminal area.

3. Fig. 5e - Esm staining increased behind the front not at the front, as stated in the text.

4. Supp Fig 5 - the parameters measured for iEC of VEGFR1 only partially overlap with iPC analysis. Highly significant changes in sprouting called "minor" when iPC mutants have similar levels of reduction and significance.

5. Supp Fig 5 - decreased branch points also not intuitive outcome from loss of EC VEGFR1 (as for PC-derived VEGFR1 earlier) - why would increased VEGF-A signaling lead to reduced branch points and filopodia?

Reviewer #2 (Remarks to the Author):

Eilken et al. report that retinal pericytes promote EC sprouting and inhibit EC proliferation through regulating VEGF-A activity in ECs. Previous in vitro studies have indicated the role of pericytes in EC growth. Here, the authors went further after they specifically depleted pericytes with genetic methods. They confirmed the role of pericytes in angiogenesis. Moreover, they found that VEGF-A downstream signals—VEGFR2, VEGFR3 and Esm1— were enriched at angiogenic blood vessels in pericyte-depleted mice. In addition, intraocular injection of VEGF-A confirmed their results from genetic ablation with DTA/DTR mice.

The results in this work are nice and convincing. The manuscript is well prepared. However, there are some concerns that should to be addressed.

1) The authors showed that Flt1iPC had similar effects as Flt1iPC/+ and concluded that the dosage of Flt1 is critical. But they didn't compare the levels of VEGFR1 in Flt1iPC and Flt1iPC/+ with western or immunostaining after injecting tamoxifen.

2) The authors showed regional increased expression of VEGFR2 in DTRiPC (Fig. 4), which was likely caused by regional loss of pericytes. In Flt1iPC/+ or Flt1iPC, the overall expression of VEGFR2 seemed increased (Fig. 6f) though the angiogenic front of ECs showed a little bit higher expression. Moreover, the increase of Esm1 was limited to the angiogenic front. How does the overall deletion of Flt1 in pericytes only affect VEGFR2/Esm1 signaling at the angiogenic front?

3) In the text (line157), the authors mentioned body weight of mice reduced following pericyte ablation. How long can these Pdgfrb-CreER;DTR mice survive after DT injection? What is the advantage of the use of DTR compared with DTA in pericyte ablation? It is not clear in the manuscript. Although the authors mentioned that Pdgfrb-creERT2;ROSA-DTA mice displayed a strong reduction in body weight, I am wondering why the authors did not inject a lower dose of tamoxifen to overcome this difficulty.

4) Pericytes have been reported important for BBB formation and maintenance. After ablation of pericytes, did the authors observe bleeding in the retina or in the brain?

Minor points

- 1) Double-check the word "Pdgfb" in the text. Sometimes it is shown as "Pdfgb".
- 2) Line 325, "intreaperitoneal" should be "intraperitoneal".
- 3) Line 230-231, the sentence "Indicating that the dosage of ...heterozygotes" is confusing.
- 4) Line 61, do vSMCs cover veins?
- 5) Line 691, delete "control (PBS)".
- 6) Line 696, add legends for Fig. 5g.
- 7) Line 662, delete "angiogenic front".
- 8) Line 273-275, the sentence "However, this conclusion..., whereas the actual function of ..." is confusing.

Reviewer #3 (Remarks to the Author):

This study addresses the role of pericytes in blood vessel function. It has been shown previously that pericytes express VEGFR1. Moreover, pericyte-deficiency has been associated with increased VEGFA production in other cell types and therefore hyperstimulation of endothelial cells. Elevated VEGFR2 signaling in turn has been shown to result in reduced branching and wider vessels. Thus the novel finding in this study is that VEGFR1 contributes to the pericyte-deficient phenotype as VEGFR1-specific ablation in pericytes recapitulates the phenotype of pericyte-deficient retinal vasculature, in particular in the angiogenic front. The study rests on a series of very elegant genetic models. Comments are given below.

1. The key question that needs to be resolved is what role VEGFR1 plays on pericytes. The authors implicate pericyte-expressed VEGFR1 in spatial restriction of VEGFA signaling. The authors state that pericyte-deficiency does not lead to changes in the VEGFA expression pattern (Fig. 4A). However, the VEGFA expression level and pattern need to be quantified, for example by RNAseq or RNAscope ISH. The expression levels and patterns of VEGFR2 and ESM1 in both pericyte-ablated and Flt1 iPC retinas also need to be quantified. It's not obvious from the images shown in Fig. 6f of a Flt1 iPC retina, that VEGFR2 is upregulated only in endothelial cells; it seems to be a broad upregulation. In contrast, ESM1 expression does not appear to be elevated in the Flt1 iPC retinas (Fig. 6h)? In conclusion, upregulation of VEGFR2 is the major consequence of complete loss of pericytes or loss of VEGFR1 expression in pericytes. The "spatial restriction" model put forward by the authors, which implies a redistribution of VEGFA from VEGFR1 to VEGFR2, is not validated. How does pericyte-expressed VEGFR1 control the expression levels of VEGFR2?

2. The Flt1 iPC retina has a surprisingly penetrant phenotype also in the het. Is it dependent on VEGFR1 signaling in pericytes or only on the potential "sink" function of VEGFR1? In Fig. 6, the authors show that pericyte coverage was not significantly reduced in the Flt1 iPC retinas, however n=3 here and perhaps performed only once? Please look at these data again since the images of desmin staining in the het and +/- floxes (Fig. 6i) are very similar to those in the pericyte-ablated images (Fig. 1c). That pericyte numbers seem affected in the Flt1 iPC clouds the interpretation of the role of VEGFR1 in pericytes. It also indicates that VEGFR1 has an important signaling function in pericytes; it's not only a sink function.

3. With regard to the pericyte quantifications: The author have now normalized pericyte numbers to blood vessel area. This is true at least for Fig. 6j while Fig. 1e needs to be better explained. Please perform the same normalization in the various experiments. However, there is a potential caveat in that vessels in the iPC are larger which means that the quantifications of the mutant samples perhaps exaggerate the relative decrease in pericyte numbers. Therefore, try to normalize to endothelial cell numbers (erg1 positive nuclei).

4. In order to clarify whether VEGFR1 has a signaling role in pericytes, the authors could e.g. isolate pericytes from wt and Flt1 iPC retinas soon after ablation and do RNAseq to determine if gene regulation is affected. Whether VEGFR1 has a gene regulatory effect or not, in pericytes, is a

very important question. The authors could also treat the Flt1 iPC mice with neutralizing VEGFA antibodies to show that liberated VEGFA is a key mechanism underlying the phenotype with wider vessels and poor sprouting.

5. It is unfortunate that developmental vascular biology studies so often are done only in the retina. After all, the retina is restricted by the blood-retina-barrier and endothelial cells have particular properties. The trachea vasculature develops through VEGF-dependent sprouting angiogenesis and like the retina, this is a 2-D tissue that can be whole-mounted (see Ni A, Lashnits E, Yao LC, Baluk P, and McDonald D M (2010). Rapid remodeling of airway vascular architecture at birth. *Dev Dyn* 239, 2354-2366.). What is the phenotype of tracheal vessel sprouting in the genetic models studied here?

First of all, we would like to thank the reviewers for their time and helpful comments, which have allowed us to improve the manuscript substantially. While we are going to provide a detailed point-by-point response to all individual questions and comments further below, we would first like to give you a summary of the most important changes and additions:

As requested by the reviewers, we have increased the amount of quantitative data on the *DTR*^{iPC} mouse model after careful measurement of the VEGF-A+ area and signal intensity (Fig. 4g), as well as VEGFR2 and VEGFR3 signal intensity (Fig. 4h and i, respectively) and Esm1+ area normalized to EC (IB4+) area (Fig. 4j). The new results obtained are in accordance with the conclusions of the original manuscript and reinforce the proposed mechanism behind pericytes regulating VEGF-induced sprouting.

We have extended our analysis of *DTA*^{iPC} mice with new data addressing the phenotype of these animals, the strong downregulation in the expression of mural cell markers (Supplementary Fig. 2f and g) and expression levels of relevant molecular players in the VEGF pathway including *Kdr*, *Vegfa*, and *Flt1* isoforms gene (Supplementary Fig. 4a, b and e). Interestingly, pericyte depletion using the *DTA*^{iPC} model, in which mural cell coverage is strongly reduced in the whole vascular plexus, triggers a more profound increase in VEGF signaling as indicated by the expression of the surrogate marker Esm1 (Supplementary Fig. 4c, d).

In order to extend the functional implications and the physiologic relevance of the findings described so far, we have analyzed the major phenotypic outcomes in the vasculature of the mouse brain cortex both in pericyte-depleted mice (*DTA*^{iPC}) and after pericyte-specific *Flt1* inactivation (*Flt1*^{iPC}). Noteworthy, in both scenarios the brain blood vessels recapitulate the most important aspects related to the deregulation of VEGF-mediated sprouting (Supplementary Fig. 4f-i and Supplementary Fig. 8).

We have performed new sets of experiments for VEGF-A intraocular injection in order to analyze ECM composition and potential changes in pericyte coverage as well as pericyte morphology and localization (Supplementary Fig. 5). Moreover, we have included additional quantitation for VEGFR2 immunosignals (Fig. 5d).

Making use of a custom-designed isoform-specific qPCR probe, we have analyzed the relative expression of soluble VEGFR1 (sFlt1) in sorted pericytes from the retina of *Hey1*-GFP reporter mice (Fig. 6b, c). Detailed confocal microscopy analysis of the P6 retinas of these mice (Fig. 6a) has allowed us to clearly show that high *Hey1* expression (bright GFP signal) constitutes a molecular marker of pericytes associated to the angiogenic front in contraposition to mural cells covering more mature vessels. Notably, these angiogenic front-associated pericytes show 3-fold enrichment in sFlt1 expression highlighting the relevance of this particular pericyte subpopulation in the control of VEGF signaling.

Moreover, we have strengthened the data regarding the *Flt1*^{iPC} model by performing additional quantitation for VEGFR2 immunosignals intensity (Fig.

7g), Esm1+ area (Fig. 7i) and by analyzing gene expression levels of VEGF ligands (Fig. 7l). Likewise, we have repeated critical stainings (Esm1) in order to better show the changes elicited after *Flt1* inactivation in pericytes (Fig. 7h). Furthermore, we have included a completely new panel (Fig. 7j) which rules out significant pericyte depletion as the cause of phenotypic alteration in the *Flt1*^{iPC} mutants. These data also clearly show that the central plexus of the retina remains unaffected, probably due to comparably low expression of soluble VEGFR1 by pericytes of this area. Finally, we have performed a series of relative gene expression analysis by qPCR to interrogate potential changes in molecules mediating EC-PC crosstalk (Supplementary Fig. 7e, f) and have included a new data point (P10) in the analysis of the *Flt1*^{iPC} phenotype (Supplementary Fig. 7g-k).

We believe that the extensive revision has substantially improved the quality of our study and we are confident that we have successfully addressed all critical concerns.

Point-by-point response:

Reviewer #1 (Remarks to the Author):

The work is timely in that, while it is clear that pericyte-endothelial cell (EC) interactions are important for vessel morphogenesis, the molecular pathways are not well-defined, and the role of VEGF-A signaling regulation from the PC compartment is not well-characterized. Enthusiasm is somewhat moderated by the focus of the work on characterization of the overall loss of PC via expression of the DTA receptor and toxin injection, relative to characterization of PC-specific loss of VEGFR1, which is considered more novel. Additionally, the work does not carefully explore (or even discuss) the effects of VEGF-A manipulation on PC-EC interactions, which would strengthen the somewhat modest conclusions.

Reply: We appreciate the critical comments and we hope that the extensive revision will satisfy the reviewer. Moreover, we would like to stress out that mural cell depletion using the DTR expression approach, rather than producing an overall loss of pericytes, is able to target preferentially pericytes associated to the angiogenic front without affecting important characteristics of the vasculature (ECM composition, junction stability, absence of a strong inflammatory response) as happens with other, more ablation approaches.

Major issues:

Reviewer #1: 1. The focus on effects of overall PC loss in post-natal retinal development lacks some novelty, as classic papers (Hellstrom et al, 2001; Lindblom et al, 2003) document reduced sprouting and hyperplasia/increased vessel diameter in mice lacking pericytes in the postnatal retina.

Reply: Only the Lindblom et al. paper from 2003¹ analyzes postnatal retina (P5) but not to great extent. In any case, it has to be noted that all model systems used by these authors (*Pdgfb* and *Pdgfrb* deficient mice, and *Pdgfb* retention motif

mutants) represent constitutive alterations in pericyte function/recruitment since early embryonic development so that potential compensatory mechanisms and indirect effects need to be considered. The models used in this manuscript are indeed novel since they allow acute genetic manipulation of pericytes in very specific time windows and therefore without affecting previous steps of development.

Reviewer #1: The focus on effects at the vascular front with somewhat cursory statements regarding effects of PC loss behind the front seems to miss an opportunity to contrast effects in angiogenic vs. remodeling/stabilized areas where PC function is better understood. Although the authors document that PC loss is not complete behind the front, some parameters (i.e. proliferation) have opposite trends depending on the location, and this is not explained or discussed.

Reply: We have focused on the angiogenic front because this is the region where phenotypic alterations are visible. This is not by accident, because, as newly added data show, pericytes at the front have distinct molecular properties (expression of the *Hey1-GFP* reporter) and express sFlt1 at much higher levels than the GFP-negative bulk of pericytes. In addition, ECs at the angiogenic front are exposed to higher levels of VEGF-A (presented by cells in the avascular retina) than the more mature endothelium of the central retina. As overly elevated VEGF-A/VEGFR2 signaling levels impair the sprouting process, the availability of VEGF-A in this region needs to be carefully balanced. Thus, pericytes at the angiogenic front are adapted to the local regulation of VEGF signaling at sprouts, which reveals functional heterogeneity of pericytes that was previously not appreciated.

Reviewer #1: This group has published beautiful work describing how trafficking that affects VEGF-A signaling is spatially localized in EC in the retina - how might this information overlap with the new PC data? As presented the new work on global PC loss seems to primarily confirm previous work.

Reply: Previous work from our lab² has shown that EC of the angiogenic front have higher rates of VEGF receptor internalization, signaling and turnover. It is therefore likely that changes in VEGF-A availability have higher impact for EC of the angiogenic front in comparison to more quiescent and mature ECs from the remodeled capillary plexus, where EC-intrinsic (cell-autonomous) mechanisms limit VEGF receptor endocytosis and signaling.

Reviewer #1: 2. The analysis of PC loss of VEGFR1 function is novel and intriguing but somewhat incomplete: a single set of data points is presented, predominantly focused on effects at the vascular front.

Reply: Following the reviewer's suggestion, we have now included P10 as an additional stage (Supplementary Fig. 7g-k) and have also extended our analysis of P6 mutants (Fig. 7f-l and Supplementary Fig. 7e, f). At P10, the formation of

deeper plexi in the retina is also compromised upon *Flt1* inactivation in pericytes despite normal mural cell coverage. Moreover, we have confirmed that no major changes occur in the remodeling plexus after *Flt1* deletion in PCs (Fig. 7j), which is the reason why we have focused on the effects at the vascular front.

Reviewer #1: The data set does not completely overlap with the data presented for intraocular VEGF-A injection, making comparisons difficult

Reply: We appreciate this comment, but it is not realistic to expect a complete overlap of the phenotypes resulting from acute manipulations (such as VEGF injection) and the much longer gene inactivation experiments. Nevertheless, major aspects of the phenotype after intraocular VEGF-A injection (reduced number of sprouts and filopodia, increased VEGFR2 intensity and Esm1+ ECs, more ECs in sprouts) are phenocopied by the loss of *Flt1* in pericytes.

Reviewer #1: 3. The concept that PC interactions with EC may be affected by any of the manipulations is not addressed. This is especially important for the VEGF manipulations that do not ablate PC. PC and EC share a basement membrane - is this intact with VEGF-A manipulation? Where are PCs relative to sprouts/filopodia at the front and does this change with VEGF manipulation?

Reply: We have repeated the VEGF-A intraocular injection experiments in order to analyze changes in the basement membrane (Supplementary Fig. 5a-c). Collagen 4 immunostaining reveals that there are no overt changes in basement membrane integrity at the angiogenic front (Supplementary Fig. 5a) despite of profound changes in sprouts morphology and capillary diameter (Supplementary Fig. 6a). Overall, the ratio of coverage of collagen 4 with respect to the IB4+ area is not altered (Supplementary Fig. 5c). We have also analyzed the expression pattern of laminin alpha 5 (Supplementary Fig. 5b), which is expressed primarily in the vascular plexus³ after induction by angiostatic factors⁴. There are no overt alterations in laminin alpha 5 at 4 hours after VEGF-A injection. In addition, we have quantified pericyte coverage after VEGF-A injection detecting only a discrete reduction. This effect is, however, not statistically significant (Supplementary Fig. 5d and 5e) and might simply reflect the enlargement/dilation of the EC area (Supplementary Fig. 5f). Finally, high magnification pictures using pericyte-specific and endothelial markers (Supplementary Fig. 5g) show that PC localization in the angiogenic front is not altered by the administration of VEGF-A.

Reviewer #1: 4. Very little discussion of the interesting data that is presented - for example, why would loss of PC decrease branch points?

Reply: Loss of pericytes affects the ability of EC to sprout and this will obviously affect the formation of new vessel interconnections and thereby branch points. However, it also needs to be taken into account that pericyte-depleted pups show a reduction in body weight that could be coupled to a delay in the

development of the retinal vasculature. Moreover, it is known that capillary branch points commonly harbor a pericyte soma and primary processes are often found to extend along each branch⁵ suggesting that pericytes may help to stabilize nascent vessel branches. We now also mention this point in the Discussion (first paragraph on page 13) of the revised manuscript.

Reviewer #1: Where do the extra EC come from 4h post-intraocular VEGF-A, since proliferation is not changed (as expected for this short time period)?

Reply: Indeed, as the reviewer points out, the increase in EC abundance in the leading angiogenic front after VEGF-A injection (Fig. 5h) is not coupled to an increase in EC proliferation (Supplementary Fig. 6c). We believe that excessive amounts of VEGF-A and the lack of a proper gradient impairs the normal cooperation of sprouting ECs similar to what has been previously proposed⁶. The resulting impaired formation of sprouts leads to the observed accumulation of ECs in enlarged vessels at the leading edge of the vascular plexus and thereby reduced extension of the vessel network.

Reviewer #1: 5. Some of the data is not quantified, and in some cases the conclusions in the paper are not well-supported by the panels shown, especially given the increased number of EC in mutants. --Fig. 4b, c, e - more EC at front in mutant, stronger if quantified and normalized to EC area

Reply: We have included quantitation of the intensity for VEGFR2 and VEGFR3 (Fig. 4h and 4i) in the angiogenic front of pericyte-depleted mice and normalized the vascular staining intensity to that of the non-vascular area (see Methods). Interestingly, the strong net increase in VEGFR2 intensity (more than two fold, Fig. 4h) exceeds the increase in EC density at the leading front (around 30%, Fig. 3d) suggesting that elevated VEGFR2 levels are not only a consequence of a higher EC number. Moreover, the higher intensity of VEGFR2 and VEGFR3 staining is not homogenous in all the sprouts or even within specific enlarged sprouts, while the increased density of ECs in the leading front vessel is rather uniform. We have also included a quantitative analysis of Esm1+ area with respect to total IB4 area (Fig. 4j), which also shows a 2-fold increase in the abundance of Esm1+ ECs reflecting elevated VEGF signaling.

Reviewer #1: --Fig. 5c - more EC at front in mutant where more R2.

Reply: Please, take into account that the VEGF-A-induced increase in VEGFR2 intensity in the vascular area is normalized to the avascular area (quantitative analysis included in this revised version, Fig. 5d and described in Methods). Moreover, the increase in VEGFR2 intensity after VEGF-A intraocular injection affects the whole vascular network and is not restricted to the angiogenic front, as is the case of the DTR mouse model. This is likely to reflect that injected VEGF-A is broadly available throughout the retinal vasculature. It also means that the increase in R2 also affects the capillary plexus even in regions without obvious

changes in EC density, which again indicates that increased R2 staining is not a consequence of more ECs. Interestingly, when we analyzed a mural cell-specific DTA mouse model in which pericyte depletion affects both the leading front and the remodeling plexus (Supplementary Fig. 2a-c), the VEGFR2 increase is also not restricted to the angiogenic front but affects the whole peripheral vasculature (Supplementary Fig. 4a, b), suggesting an important role for pericytes in controlling VEGF signaling in ECs.

Reviewer #1: --Fig. 5d, e - stated in text that more staining of ESM at front, yet graph shows no difference at front, rather more behind the front.

Reply: Esm1 is endogenously high in sprouting ECs at the angiogenic front (i.e., in ECs exposed to high VEGF) (see Fig. 5e) and it is therefore not surprising that exogenous VEGF-A, which is widely available throughout the retina, primarily affects expression behind the edge of the vascular plexus.

Reviewer #1: --Fig. 6f - R2 not quantified.

Reply: New quantitation of VEGFR2 intensity in the vasculature normalized to avascular areas shows a more than 2-fold increase in R2 intensity in the angiogenic front of *Flt1*^{iPC} mutant retinas (Fig. 7g).

Reviewer #1: --Fig. 6h - ESM not quantified and differences in mutant not apparent.

Reply: A new set of P6 pups from pericyte-specific *Flt1* mutants was perfused to eliminate unspecific background signal caused by red blood cells (see Methods), which makes it much easier to appreciate changes in Esm1 immunostaining (Fig. 7h). Quantitation of the Esm1+ area normalized to total EC area (IB4+) shows an increase of roughly 50% in the *Flt1*^{iPC} mutant angiogenic front (Fig. 7i).

Reviewer #1: --Fig. 6i - desmin seems reduced at front relative to control in presented panels.

Reply: We have carefully looked at pericyte coverage using different markers (NG2, PDGFR β , desmin) and have not detected statistically significant reduction in pericyte coverage in the retina (Fig. 7k; Supplementary Fig. 7i, k) or in the brain cortex (Supplementary Fig. 8c). The strong morphological changes in the *Flt1*^{iPC} mutant angiogenic front and the alterations in sprouts and vessel organization (which resemble the pericyte-depletion situation) might suggest reduced pericyte coverage, which is, however, not the case. In addition, it should be taken into account that certain markers (such as desmin, which was used in the previous version of the manuscript, now in Supplementary Fig. 7c) label intermediate filaments and not the whole cell body, which complicates the analysis of mural cell coverage with this particular marker.

Minor issues:

Reviewer #1: 1. Fig. 3a - here and other places the definition of the vascular front is unclear. In methods says 260 micrometers from distal most sprout, but here looks much less and the control looks more "tight" than the mutant. Since many of the measurements compare this specific area and show differences, further clarification is needed here.

Reply: We apologize for the confusion and appreciate the comment. For all quantitations using the term "angiogenic front", the approach of analyzing the first 260µm from the top of the sprouts was used. Please note that the dashed lines drawn in Fig. 3a and 5g only highlight the most distal vessels from which sprouts emerge. This issue has been clarified in the figure legends and in the Methods section.

Reviewer #1: 2. Fig. 3e - ICAM-2 does not appear apical in these panels. Suggest perfusion labeling to confirm expanded luminal area.

Reply: We have included larger magnification images (Fig. 3f), which show that the apical ICAM2 immunostaining covers a smaller area (i.e. the vessel lumen) than isolectin B4, which labels the whole endothelial cell surface.

Reviewer #1: 3. Fig. 5d - Esm staining increased behind the front not at the front, as stated in the text.

Reply: This concern was already raised in point 5 of the major issues and has been addressed.

Reviewer #1: 4. Supp Fig. 5 - the parameters measured for iEC of VEGFR1 only partially overlap with iPC analysis. Quantify number of filopodia and PC coverage in VEGFR1 iEC Highly significant changes in sprouting called "minor" when iPC mutants have similar levels of reduction and significance.

Reply: Indeed we have not measured exactly the same parameters for the *Flt1*^{iEC} mutants as for *Flt1*^{iPC} mice since the former does not show obvious changes in the angiogenic front (Fig. 7d vs. Supplementary Fig. 9f). In this regard, we have not quantified filopodia in the *Flt1*^{iEC} mutants because this parameter is not altered in the *Flt1*^{iPC} retinas, which do show a prominent defect in sprout formation. We have now included the quantitative analysis for pericyte coverage in *Flt1*^{iEC} mutants (Supplementary Fig. 9c, d) showing no significant changes. With respect to the alterations in the number of sprouts comparing endothelial and pericyte-specific mutants, we disagree. In iPC mutants, the number of sprouts shows a 40% reduction (vs. 15% in iEC mutants) and a p-value of 0.0001 (vs. 0.0165 in iEC). Again, this difference is already very obvious when one looks

at the vessel front in the two mutant models (Fig. 7d vs. Supplementary Fig. 9d).

Reviewer #1: 5. Supp Fig. 5 - decreased branch points also not intuitive outcome from loss of EC VEGFR1 (as for PC-derived VEGFR1 earlier) - why would increased VEGF-A signaling lead to reduced branch points and filopodia?

Reply: As mentioned before, a reduction of sprouting would also lead to a smaller number of new anastomotic connections and thereby a reduction in branch points. Aberrantly elevated VEGF-A/VEGFR2 signaling is known to lead to reduced sprouting (see, for example, Fig. 7g, k in Gerhardt *et al.*⁷ or Bentley & Chakravartula, 2017⁶).

Reviewer #2 (Remarks to the Author): Eilken *et al.* report that retinal pericytes promote EC sprouting and inhibit EC proliferation through regulating VEGF-A activity in ECs. Previous *in vitro* studies have indicated the role of pericytes in EC growth. Here, the authors went further after they specifically depleted pericytes with genetic methods. They confirmed the role of pericytes in angiogenesis. Moreover, they found that VEGF-A downstream signals (VEGFR2, VEGFR3 and Esm1) were enriched at angiogenic blood vessels in pericyte-depleted mice. In addition, intraocular injection of VEGF-A confirmed their results from genetic ablation with DTA/DTR mice.

The results in this work are nice and convincing. The manuscript is well prepared. However, there are some concerns that should be addressed.

Reply: We are grateful for the positive assessment of our work and the helpful feedback.

Reviewer #2: 1) The authors showed that Flt1iPC had similar effects as Flt1iPC/+ and concluded that the dosage of Flt1 is critical. But they didn't compare the levels of VEGFR1 in Flt1iPC and Flt1iPC/+ with western or immunostaining after injecting tamoxifen.

Reply: We think that this experiment is technically not feasible because it would require a detectable change in Flt1/sFlt1 expression in whole retina lysates. New data (Fig. 6a-c) indicates that we have local differences in sFlt1 expression by pericytes. Whereas *Hey1*-GFP⁺ pericytes at the angiogenic front show enrichment of sFlt1 transcripts relative to input, the same is not the case for the bulk of GFP-negative pericytes. Thus, we are looking at a local function of pericyte-derived sFlt1 at the angiogenic front. In addition, there is substantial Flt1/sFlt1 expression by endothelial cells and perhaps other cell populations (such as microglia), which should make it impossible to see significant alterations in total retina lysates.

Reviewer #2: 2) The authors showed regional increased expression of VEGFR2 in DTRiPC (Fig. 4), which was likely caused by regional loss of pericytes. In Flt1iPC/+ or Flt1iPC, the overall expression of VEGFR2 seemed increased (Fig. 6f) though the angiogenic front of ECs showed a little bit higher expression. Moreover, the increase of Esm1 was limited to the angiogenic front. How does the overall deletion of Flt1 in pericytes only affect VEGFR2/Esm1 signaling at the angiogenic front?

Reply: On one hand, the angiogenic front is the region that is most exposed to the VEGF-A gradient (see Fig. 4a), which is also reflected in the local expression of markers such as Esm1 (Fig. 4e). This points out that VEGF-A/VEGFR2 expression and activity need to be carefully controlled especially in this area. Our new data also suggests that sFlt1 expression is a feature of a specific (*Hey1-GFP+*) pericyte subset at the angiogenic front. Both aspects together can readily explain why we see profound defects at the edge of the growing vessel plexus (Fig. 7d), whereas the vasculature of the central retina appears unaffected (Fig. 7j).

Reviewer #2: 3) In the text (line157), the authors mentioned body weight of mice reduced following pericyte ablation. How long can these *Pdgfrb-CreER;DTR* mice survive after DT injection? What is the advantage of the use of DTR compared with DTA in pericyte ablation? It is not clear in the manuscript. Although the authors mentioned that *Pdgfrb-creERT2;ROSA-DTA* mice displayed a strong reduction in body weight, I am wondering why the authors did not inject a lower dose of tamoxifen to overcome this difficulty.

Reply: Indeed, diphtheria toxin/DTR-mediated pericyte ablation affects the body weight of the pups by day 6 of postnatal development. Nevertheless, there is not a notorious change in survival in comparison to control littermates injected with tamoxifen and diphtheria toxin. The key advantage of the DTR approach is that the extent of pericyte ablation can be more carefully controlled through the dosage of the administered diphtheria toxin and, in the particular case of the retina, effects are primarily confined to pericytes associated with the angiogenic front. This is now clearly stated in the manuscript. In contrast, it is much more difficult to control the dosage of tamoxifen or 4-OHT-induced diphtheria toxin expression in the DTA model. Because of these relevant differences, we have chosen the DTR model to avoid the strong inflammatory response, vessel leakage, rapid loss of body weight and lethality associated with *Pdgfrb-CreERT2*-controlled DTA expression. Moreover, we have expanded the analysis in the DTA mouse model, where the pericyte depletion affects the whole retinal vasculature bed more homogeneously (Supplementary Fig. 2 and 3).

Reviewer #2: 4) Pericytes have been reported important for BBB formation and maintenance. After ablation of pericytes, did the authors observe bleeding in the retina or in the brain?

Reply: Only after pericyte ablation using the DTA model it was possible to detect vessel leakage and occasional bleedings in the retina, although we did not observe hemorrhages in the brain. The DTR approach yielded no detectable signs of vessel leakage or bleeding in any region of the central nervous system including the retina. Moreover, endothelial cell junctions were intact (Supplementary Fig. 3d-f).

Minor points

Reviewer #2: 1) Double-check the word Pdgfb; in the text. Sometimes it is shown as Pdfgb;. 2) Line 325, intraperitoneal; should be intraperitoneal.

Reply: Thank you for alerting us to these mistakes, which have been corrected.

Reviewer #2: 3) Line 230-231, the sentence “Indicating that the dosage of heterozygotes” is confusing.

Reply: We have slightly rephrased this sentence and think that the message is clear.

Reviewer #2: 4) Line 61, do vSMCs cover veins?

Reply: vSMCs are present in both arteries and veins although they differ in their shape and degree of coverage. In normal development, the arterial coverage can be seen earlier.

Reviewer #2: 5) Line 691, “control (PBS)”

6) Line 696, add legends for Fig. 5g.
7) Line 662, “angiogenic front”

Reply: Thank you for alerting us to these mistakes, which have been corrected.

Reviewer #2: 8) Line 273-275, the sentence “However, this conclusion..., whereas the actual function of” is confusing.

Reply: In our view, this sentence is clear and correct because the role of angiopoietin-1 expression by pericytes has not been investigated directly *in vivo*. Global mutant mice or pharmacological manipulation are obviously not cell type-specific.

Reviewer #3 (Remarks to the Author):

This study addresses the role of pericytes in blood vessel function. It has been shown previously that pericytes express VEGFR1. Moreover, pericyte-deficiency

has been associated with increased VEGFA production in other cell types and therefore hyperstimulation of endothelial cells. Elevated VEGFR2 signaling in turn has been shown to result in reduced branching and wider vessels. Thus the novel finding in this study is that VEGFR1 contributes to the pericyte-deficient phenotype as VEGFR1-specific ablation in pericytes recapitulates the phenotype of pericyte-deficient retinal vasculature, in particular in the angiogenic front. The study rests on a series of very elegant genetic models. Comments are given below.

Reply: We are grateful for this assessment and have improved our study by adding new data on pericyte heterogeneity and sFlt1 expression in the developing retina.

Reviewer #3: 1. The key question that needs to be resolved is what role VEGFR1 plays on pericytes. The authors implicate pericyte-expressed VEGFR1 in spatial restriction of VEGFA signaling. The authors state that pericyte-deficiency does not lead to changes in the VEGFA expression pattern (Fig. 4A). However, the VEGFA expression level and pattern need to be quantified, for example by RNAseq or RNAscope ISH

Reply: We show that *Vegfa* expression is strongly elevated after pericyte ablation in the DTA model (Supplementary 4e), which is consistent with similar findings in recent studies from the Koh⁸ and Uemura⁹ laboratories. Both studies also show that very stringent pericyte ablation leads to breakdown of the blood-retina barrier and profound inflammation. In our DTR model, the more limited reduction of pericytes avoids the upregulation of *Vegfa*. We have documented this both by quantitative analysis of VEGF-A positive area and signal intensity in immunostained retinas (Fig. 4a, g), and by qPCR analysis of VEGF-A expression in whole retina lysates (Fig. 7l). The angiogenic front-targeted depletion of pericytes elicited by the *DTR*^{IPC} mouse model thereby enables new interesting insights into the function of pericytes.

Reviewer #3: The expression levels and patterns of VEGFR2 and ESM1 in both pericyte-ablated and Flt1 IPC retinas also need to be quantified.

Reply: As requested by the reviewer, we have added quantitations for the DTR (Fig. 4h, j) and DTA pericyte-depletion models (Supplementary Fig. 4b, d) as well as for the *Flt1*^{IPC} mutants (Fig. 7g, i).

Reviewer #3: It's not obvious from the images shown in Fig. 6f of a Flt1 IPC retina, that VEGFR2 is upregulated only in endothelial cells; it seems to be a broad upregulation.

Reply: Quantitation of VEGFR2 intensity in the vasculature (Fig. 7g) involves subtracting the intensity of the non-vascular area in each image analyzed (see Methods). This normalization step accounts for sample-specific variations in antibody staining and rules out the potential contribution of background signal.

There is a discrete increase (21.6%) in the staining intensity of non-vascular structures in *Flt1*^{iPC} retinas, which is, however, not statistically significant. In contrast, the difference between vascular and non-vascular tissues is greatly increased (2.3 fold, p=0.0164) in mutant retinas.

Reviewer #3: In contrast, ESM1 expression does not appear to be elevated in the Flt1 iPC retinas (Fig. 6h)?

Reply: The staining was repeated using retinas from animals that were perfused with PBS through the left ventricle (see Methods) in order to wash out serum and red blood cells from the blood vessels, which were giving some unspecific background signal that obscured some of the differences in the previous Esm1 stainings. We have therefore replaced the Esm1 image in Fig. 7h and performed quantitation of the Esm1 area in the angiogenic front normalized to the whole vasculature (IB4+) of the same area. Together, these data clearly show upregulation of Esm1 in *Flt1*^{iPC} retinas.

Reviewer #3: In conclusion, upregulation of VEGFR2 is the major consequence of complete loss of pericytes or loss of VEGFR1 expression in pericytes.

Reply: A complete or nearly complete loss of pericytes leads to profound *Vegfa* upregulation and vessel leakage. More limited manipulation, such as pericyte ablation with the DTR model or in *Flt1*^{iPC} mutants, lead to local increase in VEGF-A/VEGFR2 signaling (visible as increased Esm1 expression), which we attribute to locally enhanced VEGF-A bioavailability.

Reviewer #3: The “spatial restriction”; model put forward by the authors, which implies a redistribution of VEGFA from VEGFR1 to VEGFR2, is not validated. How does pericyte-expressed VEGFR1 control the expression levels of VEGFR2?

Reply: Upregulated VEGFR2 and Esm1 expression are consequences of enhanced VEGF-A bioavailability and VEGF-A/VEGFR2 signaling, as the intraocular injection experiments show (Fig. 5c-f). Pericytes and, in particular, the *Hey1-GFP*⁺ subset at the angiogenic front express sFlt1, which locally restricts the distribution/availability of VEGF-A at the edge of the growing retinal vasculature.

Reviewer #3: 2. The Flt1 iPC retina has a surprisingly penetrant phenotype also in the het. Is it dependent on VEGFR1 signaling in pericytes or only on the potential “sink” function of VEGFR1?

Reply: We cannot (and do not want to) rule out a potential signaling role of VEGFR1 in pericytes. However, this interesting question is not within the scope of the current manuscript. In addition, we also lack obvious pericyte defects in *Flt1*^{iPC} mutants, which do not show significant changes in pericyte coverage (Fig.

7k) or overt alterations in marker expression (Fig. 7h, j and Supplementary Fig. 7c).

Reviewer #3: In Fig. 6, the authors show that pericyte coverage was not significantly reduced in the Flt1 iPC retinas, however n=3 here and perhaps performed only once? Please look at these data again since the images of desmin staining in the het and -/- floxes (Fig. 6i) are very similar to those in the pericyte-ablated images (Fig. 1c) That pericyte numbers seem affected in the Flt1 iPC clouds the interpretation of the role of VEGFR1 in pericytes. It also indicates that VEGFR1 has an important signaling function in pericytes; it's not only a sink function.

Reply: We have carefully checked this data and repeated the analysis using additional pericyte markers. The quantitation of pericyte coverage in the new data set (Fig. 7k) is consistent with the previous results (now in Supplementary Fig. 7d), which show a minor but not significant reduction in pericyte coverage. As desmin does mark intermediate filaments and not the whole cell body and its projections, we are also showing NG2 immunostaining in Fig. 7j. With regard to a potential signaling function of VEGFR1 in pericytes, we have addressed this point above.

Reviewer #3: 3. With regard to the pericyte quantifications: The authors have now normalized pericyte numbers to blood vessel area. This is true at least for Fig. 6j while Fig. 1e needs to be better explained. Please perform the same normalization in the various experiments. However, there is a potential caveat in that vessels in the iPC are larger which means that the quantifications of the mutant samples perhaps exaggerate the relative decrease in pericyte numbers. Therefore, try to normalize to endothelial cell numbers (erg1 positive nuclei).

Reply: Whenever pericyte coverage is quantified in the manuscript, the data has been normalized to the blood vessel area. We agree with the caveat pointed out by the referee but disagree with the suggested solution. As pericyte coverage refers to the amount of blood vessel surface covered by pericyte cell bodies, it is not obvious how the analysis of EC nuclei and thereby EC number would change the conclusions.

Reviewer #3: 4. In order to clarify whether VEGFR1 has a signaling role in pericytes, the authors could e.g. isolate pericytes from wt and Flt1 iPC retinas soon after ablation and do RNAseq to determine if gene regulation is affected. Whether VEGFR1 has a gene regulatory effect or not, in pericytes, is a very important question.

Reply: We agree with the reviewer regarding the potential importance of this question, but we think that this approach would require a separate study. In addition, we may not detect any changes by analyzing the whole pericyte population and, instead, it might be necessary to isolate distinct pericyte subsets

or perform single cell RNA sequencing. In the current manuscript, we have analyzed the expression of several molecules involved in EC-pericyte communication in whole retina lysates without detecting significant changes. We are well aware of the limitations of this approach, but we have seen in some of our other projects that pericyte-specific alterations can lead to changes in signaling that are detectable at the whole organ level.

Reviewer #3: The authors could also treat the Flt1 iPC mice with neutralizing VEGFA antibodies to show that liberated VEGFA is a key mechanism underlying the phenotype with wider vessels and poor sprouting.

Reply: We know from our own work that the injection of blocking antibodies against VEGF-A leads to the loss of Esm1 expression, strongly alters vessel morphology (loss of branch points and induction of vessel dilation) and also impairs sprouting. The latter automatically implies that we cannot expect to rescue morphological defects in *Flt1*^{iPC} mutants by blocking VEGF. It is obviously critical to have the right level of VEGF-A expression.

Reviewer #3: 5. It is unfortunate that developmental vascular biology studies so often are done only in the retina. After all, the retina is restricted by the blood-retina-barrier and endothelial cells have particular properties. The trachea vasculature develops through VEGF-dependent sprouting angiogenesis and like the retina, this is a 2-D tissue that can be whole-mounted (see Ni A, Lashnits E, Yao LC, Baluk P, and McDonald D M (2010). Rapid remodeling of airway vascular architecture at birth. *Dev Dyn* 239, 2354-2366.). What is the phenotype of tracheal vessel sprouting in the genetic models studied here?

Reply: We appreciate the opinion of the reviewer, but we also think that there is benefit in the fact that a lot of functional studies use the same model system so that different findings can be compared enabling the integration of different findings.

To address the reviewer's concern, we have now validated some of our findings in the brain cortex of DTA (Supplementary Fig. 4f-i) and *Flt1*^{iPC} mutants Flt1 mutants (Supplementary Fig. 8a-j). Similar to the retinal phenotype, pericyte depletion or pericyte-specific *Flt1* inactivation induces a strong increase in VEGFR2 immunostaining in the brain cortex without affecting pericyte coverage. Moreover, lack of VEGFR1 in brain pericytes reduces the amount of sprouts in the brain cortex suggesting that our findings in retina also apply to the developing brain vasculature.

References for responses to comments of the reviewers

- 1 Lindblom, P. *et al.* Endothelial PDGF-B retention is required for proper investment of pericytes in the microvessel wall. *Genes Dev* **17**, 1835-1840, doi:10.1101/gad.266803 (2003).
- 2 Nakayama, M. *et al.* Spatial regulation of VEGF receptor endocytosis in angiogenesis. *Nat Cell Biol* **15**, 249-260, doi:10.1038/ncb2679 (2013).

- 3 Stenzel, D. *et al.* Endothelial basement membrane limits tip cell formation
by inducing Dll4/Notch signalling in vivo. *EMBO Rep* **12**, 1135-1143,
doi:10.1038/embor.2011.194 (2011).
- 4 Yousif, L. F., Di Russo, J. & Sorokin, L. Laminin isoforms in endothelial and
perivascular basement membranes. *Cell Adh Migr* **7**, 101-110,
doi:10.4161/cam.22680 (2013).
- 5 Armulik, A., Genove, G. & Betsholtz, C. Pericytes: developmental,
physiological, and pathological perspectives, problems, and promises. *Dev*
Cell **21**, 193-215, doi:10.1016/j.devcel.2011.07.001 (2011).
- 6 Bentley, K. & Chakravartula, S. The temporal basis of angiogenesis. *Philos*
Trans R Soc Lond B Biol Sci **372**, doi:10.1098/rstb.2015.0522 (2017).
- 7 Gerhardt, H. *et al.* VEGF guides angiogenic sprouting utilizing endothelial
tip cell filopodia. *J Cell Biol* **161**, 1163-1177, doi:10.1083/jcb.200302047
jcb.200302047 [pii] (2003).
- 8 Park, D. Y. *et al.* Plastic roles of pericytes in the blood-retinal barrier. *Nat*
Commun **8**, 15296, doi:10.1038/ncomms15296 (2017).
- 9 Ogura, S. *et al.* Sustained inflammation after pericyte depletion induces
irreversible blood-retina barrier breakdown. *JCI Insight* **2**, e90905,
doi:10.1172/jci.insight.90905 (2017).

Reviewers' comments:

Reviewer #1 (Remarks to the Author):

This revised paper by Eiken et al. is significantly improved, with additional quantification of data, and additional experiments that by and large support the conclusions. There are a few remaining concerns, mostly regarding interpretation of the magnitude of the effect of PC loss of Flt1 (and relative to EC loss):

1. A question posed as to how loss of PC Flt1 could result in up-regulation of Flk1 expression as documented (presumably in EC) was not adequately answered by the authors.

2. The difference in sFlt expression (about 3 fold) between the Hey-GFP+ and Hey-GFP neg cells is modest, especially relative to the large (almost 400 fold) enrichment for pdgfrb in the same preps. Since all was normalized to control it suggests that sFlt expression is not robust in PC, even in the sub-population at the front.

3. PC ablation via DTA did not change the overall amount of sFlt in retinal lysates, again suggesting that the bulk of sFlt is produced outside the PC compartment. It would be useful to have a better sense of relative levels of sFlt in the different cell types.

4. The changes in vessel area, branching and sprouting are significant with loss of EC Flt1. The authors argue that they are 'minor' relative to PC loss especially for sprouting by comparing 40% reduction to 15% reduction and citing p-values. It is not appropriate to make those quantitative comparisons across distinct experiments with numerous variables; the changes are either significant or not. And the p-values for control vs. Flt1PC (0.0121) are very close to the p-value for control vs. Flt1EC (0.0165). Text edits are recommended to better reflect that both manipulations affect vessel morphogenesis.

Reviewer #2 (Remarks to the Author):

The revised manuscript has been much improved. The authors have fully addressed my concern. I don't have further question about this work.

Reviewer #3 (Remarks to the Author):

In the revised manuscript by Eilken et al. the authors have responded to most if not all of my concerns. The finding that pericytes impact endothelial cells primarily by VEGFR1-regulated VEGFA bioavailability is novel and interesting. Using elegant genetic models the authors have managed to single out the critical role of VEGFR1 expression without interfering with other aspects of pericyte function, and moreover, created a condition of acute pericyte-deficiency mimicking the VEGFR1-specific loss. Even though the critical question of the role of VEGFR1 signalling in pericytes remains and is important, overall the data agree with a primary role for pericyte-expressed VEGFR1 in restricting VEGFA bioavailability.

Reviewer #1 (Remarks to the Author):

This revised paper by Eiken et al. is significantly improved, with additional quantification of data, and additional experiments that by and large support the conclusions. There are a few remaining concerns, mostly regarding interpretation of the magnitude of the effect of PC loss of Flt1 (and relative to EC loss):

Reviewer #1: 1. A question posed as to how loss of PC Flt1 could result in up-regulation of Flk1 expression as documented (presumably in EC) was not adequately answered by the authors.

Reply: Our data indicates that pericyte-derived VEGFR1 restricts local VEGF-A bioavailability particularly at the angiogenic front. In the absence of pericytes, of pericyte-derived VEGFR1 or after injection of VEGF-A, a clear upregulation of VEGFR2 can be observed, which we see as a readout of enhanced VEGFR2 signaling. This notion is now further supported by new *in vitro* experiments (included in Supplementary Figure 5h), which show VEGF-A treatment increases *KDR* gene transcription in cultured endothelial cells (HUVECs) almost two-fold in comparison to control (vehicle-treated) cells over a period lasting at least 4 days. These results are in line with a previous report¹, which concluded that VEGF is able to upregulate expression of its own receptor after activation of the VEGFR2 tyrosine kinase. Moreover, we also show *that in vitro* VEGF-A stimulation of endothelial cells is able to drive a statistically significant increase in *ESM1* gene expression (Supplementary Figure 5i), which recapitulates our *in vivo* observations.

Reviewer #1: 2. The difference in sFlt expression (about 3 fold) between the Hey-GFP+ and Hey-GFP neg cells is modest, especially relative to the large (almost 400 fold) enrichment for *pdgfrb* in the same preps. Since all was normalized to control it suggests that sFlt expression is not robust in PC, even in the sub-population at the front.

Reply: It is, in our view, probably misleading to compare expression levels of sFlt1 and PDGFR β , as the latter is a rather specific marker for mural cells in the retina, whereas sFlt1 is obviously expressed by multiple cell types (including ECs). This aspect is critical because normalization of gene expression was done relative to the signal seen in whole retina cell suspensions. Comparing Flt1 expression in ECs with the one in pericytes is further complicated by the fact that the strongest immunosignals for Flt1 can be seen in arteries (Flt1 is also frequently used as an artery marker in zebrafish). Thus, the strongest site of endothelial Flt1 expression is relatively far away from the angiogenic growth front and sprouts. In contrast, Hey1-GFP+ pericytes are found in close proximity of endothelial sprouts (see Fig. 6a), which is consistent with the local phenotype seen after *Pdgfrb-CreERT2*-mediated *Flt1* inactivation. Thus, *Flt1* expression in pericytes is biologically relevant irrespective of other expression sites and irrespective of the relative levels expressed by different cell populations.

Reviewer #1: 3. PC ablation via DTA did not change the overall amount of sFlt in retinal lysates, again suggesting that the bulk of sFlt is produced outside the PC compartment. It would be useful to have a better sense of relative levels of sFlt in the different cell types.

Reply: We agree with the reviewer that this is an important point to clarify and therefore we have performed a direct analysis of VEGFR1 expression in the retina using a rat monoclonal antibody (ImClone Systems, clone MF1). The results are summarized in the figure provided further below.

Anti-VEGFR1 immunostaining confirms strong expression of the receptor in blood vessels, as was previously described by various groups²⁻⁶. Other sources of VEGFR1 may include the retinal nerve fiber layer². Within the retinal vasculature, expression is highest in arteries and shows a sharp reduction in capillaries emerging from the distal arteries. Lower expression is seen in the angiogenic front (particularly in tip cells) and in veins (Revision Figure 1a). Similar regional differences in the level of expression of VEGFR1 were proposed by Matsumoto *et al.*³, who reported that Flt1 expression levels in retinal stalk cells and in endothelial cells from large blood vessels are consistently higher than those in tip cells.

We also provide data below (Revision Figure 1c), which shows that total Flt1 expression in the endothelium is significantly higher than in pericytes. However, as mentioned already above, endothelial expression is highest in arteries, whereas signals at the front are comparably low.

In line with the Hey1-GFP reporter data (shown in Fig. 6), there is also a subset of pericytes near the angiogenic front that shows detectable levels of VEGFR1 (Revision Figure 1b). This may indicate that most other pericytes do not express VEGFR1 or produce predominantly the soluble isoform, which is presumably not detectable by immunostaining (as is the case for most secreted proteins).

Altogether, we do not argue that the bulk of Flt1 may be produced outside the pericyte compartment, but, in our view, it is clear that pericyte-derived VEGFR1 is required for the regulation of sprouting angiogenesis. In addition, endothelial VEGFR1 expression cannot compensate for the loss of *Flt1* in pericytes.

Revision Figure 1. VEGFR1 expression in the P6 retinal vasculature.

a, Confocal images of P6 *Flt1*^{iEC} and control retinas stained for IB4 (red) and VEGFR1 (white). Note the strong reduction in VEGFR1 intensity in the endothelial-specific *Flt1* knockout (*Flt1*^{iEC}) reflecting antibody specificity. VEGFR1 expression is shown in control arteries (a), distal arterioles (da). Lower expression can be seen in the capillary plexus of the angiogenic front (AF) and in veins (v). Scale bar, 50 μ m.

b, Confocal images showing VEGFR1 (white) expression in endothelial cells (IB4, red) or mural cells (NG2, green). Images on last column show higher magnification of boxed insets. Note that VEGFR1 is barely or not detectable in the majority of mural cells (white arrowheads) with the exception of some pericytes in the angiogenic front (yellow arrowhead). Scale bar, 50 μ m.

c, RNA-sequencing analysis of *Flt1* expression in P6 *Rpl22*^{tm1.Psam} transgenic mice bred to *Pdgfrb-CreERT2* (mural) or *PdgfbCre*⁷ (endothelial) Cre-driver lines.

Reviewer #1: 4. The changes in vessel area, branching and sprouting are significant with loss of EC Flt1. The authors argue that they are 'minor' relative to PC loss especially for sprouting by comparing 40% reduction to 15% reduction and citing p-values. It is not appropriate to make those quantitative comparisons across distinct experiments with numerous variables; the changes are either significant or not. And the p-values for control vs. FltiPC (0.0121) are very close to the p-value for control vs. FltiEC (0.0165). Text edits are recommended to better reflect that both manipulations affect vessel morphogenesis.

Reply: We do not intend to challenge published studies showing that endothelial Flt1 controls EC behavior in sprouting angiogenesis. This point was already mentioned in the Discussion of the previous version (last paragraph on page 14). Moreover, we agree with the reviewer that the comparison of phenotypic outcomes across different mouse mutants is not trivial. We have revised the Results (2nd paragraph on page 12) to emphasize that EC-specific Flt1 mutants “displayed statistically significant defects in vessel branching, vascular area, sprout formation and EC proliferation.”

The other statements, namely the different morphologies of endothelial sprouts in EC-specific and pericyte-specific Flt1 mutants as well as the fact that EC-derived Flt1 does not compensate for the inactivation of the gene in pericytes, are fully supported by the data presented in the manuscript.

References for responses to comments by Reviewer #1

- 1 Shen, B. Q. *et al.* Homologous up-regulation of KDR/Flk-1 receptor expression by vascular endothelial growth factor in vitro. *J Biol Chem* **273**, 29979-29985 (1998).
- 2 Gariano, R. F., Hu, D. & Helms, J. Expression of angiogenesis-related genes during retinal development. *Gene Expr Patterns* **6**, 187-192, doi:10.1016/j.modgep.2005.06.008 (2006).
- 3 Matsumoto, K. *et al.* Study of normal and pathological blood vessel morphogenesis in Flt1-tdsRed BAC Tg mice. *Genesis* **50**, 561-571, doi:10.1002/dvg.22031 (2012).
- 4 Minami, T., Donovan, D. J., Tsai, J. C., Rosenberg, R. D. & Aird, W. C. Differential regulation of the von Willebrand factor and Flt-1 promoters in the endothelium of hypoxanthine phosphoribosyltransferase-targeted mice. *Blood* **100**, 4019-4025, doi:10.1182/blood-2002-03-0955 (2002).
- 5 Okabe, K. *et al.* Neurons limit angiogenesis by titrating VEGF in retina. *Cell* **159**, 584-596, doi:10.1016/j.cell.2014.09.025 (2014).
- 6 Shih, S. C., Ju, M., Liu, N. & Smith, L. E. Selective stimulation of VEGFR-1 prevents oxygen-induced retinal vascular degeneration in retinopathy of prematurity. *J Clin Invest* **112**, 50-57, doi:10.1172/JCI17808 (2003).
- 7 Claxton, S. *et al.* Efficient, inducible Cre-recombinase activation in vascular endothelium. *Genesis* **46**, 74-80, doi:10.1002/dvg.20367 (2008).